# Quantifying the feedback between Antarctic meltwater release and subsurface Southern Ocean warming

Erwin Lambert[1], Dewi Le Bars[1], Eveline van der Linden[1], André Jüling[1], and Sybren Drijfhout[1,2]

[1]Royal Netherlands Meteorological Institute (KNMI), Utrechtseweg 297, 3731 GA, De Bilt, The Netherlands
[2]Institute for Marine and Atmospheric Research Utrecht, Department of Physics, Utrecht University, Princetonplein 5, 3584 CC, Utrecht, The Netherlands

**Correspondence:** erwin.lambert@knmi.nl

**Abstract.** The subsurface ocean around Antarctica is one of the primary drivers of mass loss from the Antarctic ice sheet through the basal melting of ice shelves. The resultant meltwater flux into the surrounding ocean can, mainly through reduced vertical mixing, further enhance subsurface ocean warming, inducing a positive feedback that amplifies mass loss and sea-level rise. This feedback is omitted in most sea-level projections though, as few Earth System Models are fully coupled to an interactive model of the Antarctic ice sheet. Here, we quantify this feedback between Antarctic meltwater release and ocean warming using Linear Response Functions in the Earth System Model EC-Earth3. Increased meltwater release from five individual Antarctic ice sheet regions is found to unambiguously warm the subsurface Southern Ocean at centennial time scales in EC-Earth3. This warming response is quantified in terms of Linear Ocean Response Functions. Combining these with Linear Response Functions of the Antarctic mass loss and sea level rise because of ocean warming, allows for the quantification of the meltwater–ocean-warming feedback. Here, this feedback is calculated for ocean temperature projections from 14 CMIP6 Earth System Models, and Linear Response Functions from 8 Ice Sheet Models. Using a fixed basal melt relation to ocean temperatures, the feedback enhances 21st century projections of the Antarctic sea-level contribution by approximately 80%. However, the inclusion of this feedback necessitates a calibration of the basal melt relation in order to reproduce historical ice-mass loss. This calibration leads to a reduction in the basal melt parameter by 35%, after which the 21st century sea-level enhancement due to the feedback decreases to a mere 5%. We propose that a similar reduction in the basal melt parameter must be applied in ice-sheet model forcing when transitioning from a stand-alone setup to an ice-sheet–ocean coupled setup in which the meltwater–warming feedback is explicitly simulated.

## 1  Introduction

Despite substantial progress in ice-sheet modelling (e.g., Goelzer et al., 2020; Seroussi et al., 2020), uncertainties in projections of future Antarctic mass loss appear to have increased over time (Bamber et al., 2019; Levermann et al., 2020; Edwards et al., 2021). One key reason for this is the absence of a fully interactive ice sheet model in most Earth System Models (ESMs). The lack of fully coupled ice-sheet–climate simulations prevents the explicit computation of ice-sheet–ocean feedbacks in simulations of Antarctic mass loss. In this study, we present an alternative approach to explicit numerical modelling to quantify this feedback and its impact on sea-level projections.

Two mechanisms exist that induce a positive feedback between meltwater release and ocean temperatures. The first is the buoyancy-driven circulation (Jenkins, 1991). Subsurface meltwater release, due to ice shelf basal melt, induces an overturning circulation within ice shelf cavities. This circulation pulls in relatively warm offshore water masses. This circulation-induced feedback is typically accounted for in ice sheet modelling by a quadratic basal melt dependence on subsurface ocean temperatures (Favier et al., 2019; Jourdain et al., 2020). The second mechanism is a large-scale stratification change. As meltwater rises to the surface around the continent, it increases the salinity-dominated stratification and thereby suppresses deep convection (Fogwill et al., 2015). This in turn reduces the upward heat transfer, leading to a potential convergence of subsurface heat. This second mechanism is currently not accounted for in ice sheet modelling studies. It is this stratification-induced feedback which we quantify in the current study.

The stratification-induced warming of the subsurface Southern Ocean has been confirmed by both observations and simulations. Observations in the Southern Ocean reveal a number of multi-decadal trends, including cooling and freshening in the upper 200 m and warming below 200 m (Haumann et al., 2020; Auger et al., 2021). Such warming was expected based on modelling studies (Hansen et al., 2016; Pauling et al., 2016) and the observed trends have been attributed to increased meltwater release and upper-ocean stratification by modelling studies (e.g., Li et al., 2023). The regional extent of this warming impact, however, has been subject of dispute between modelling studies. According to model simulations by Golledge et al. (2019), this stratification-induced feedback mainly enhances subsurface warming in the Amundsen region. However, according to high-resolution 10-year simulations by Moorman et al. (2020), meltwater release reduces warming in the Amundsen region and increases it in the Ross region through changes in the current structure on the shelf. This result is qualitatively confirmed with a coarser resolution ESM (Thomas et al., 2023). With the exception of the Amundsen Sea, modelling studies generally agree on a subsurface warming effect, hence confirming that the stratification-induced feedback is on average a positive one with the potential to enhance ice shelf basal melt. Recently, the SOFIA (Southern Ocean Freshwater Input from Antarctica) Initiative was initialised (Swart et al., 2023) in order to assess the inter-model differences in the sensitivity to Southern Ocean freshwater input.

The explicit modelling of the stratification-induced feedback between meltwater release and subsurface ocean warming would require the coupling of ESMs with a fully interactive ice sheet model. Such developments are still in their infancy (Smith et al., 2021; Pelletier et al., 2022). Coupling efforts are impeded by large biases in ESM variables used to force the ice sheet model (Seroussi et al., 2020; Gorte et al., 2020); a highly uncertain relation between ocean conditions and ice shelf basal melt (Jourdain et al., 2020); and a strong sensitivity of ice sheet models to initial conditions (Seroussi et al., 2019). Because of these difficulties, projections of Antarctic ice mass loss are based on stand-alone simulations forced with anomalies from ESM simulations, as in the Ice Sheet Model Intercomparison Project (ISMIP)6 (Nowicki et al., 2020; Seroussi et al., 2020). Comparing these projections to recent observations of mass loss revealed a great model spread, indicating an inconsistency between models and observations (Aschwanden et al., 2021). One possible reason for this inconsistency is the lack of a representation of the feedback between meltwater and ocean warming in stand-alone simulations.

As an alternative to explicit numerical modelling, some studies have used Linear Response Functions (LRFs). These functions are mathematical approximations to the response of a dynamical system to a change in external forcing, derived from

perturbation studies. LRFs have been derived for the response of ice sheet models to an increase in basal melt (LARMIP-2, Levermann et al., 2020). In that study, ice shelf basal melt was parameterised as a linear function of subsurface ocean temperatures around the Antarctic continent, derived from ESM simulations. This relation was improved by van der Linden et al. (2023), who calibrated it to reproduce historical observed ice mass loss from Rignot et al. (2019).

In this study, we build upon the LRF framework and extend it to quantify the feedback between subsurface ocean warming and ice shelf basal melt. Using simulations with the ESM EC-Earth3 (Döscher et al., 2022), we construct response functions for the response of subsurface ocean temperatures to an increase in meltwater release. We then combine these response functions with the response functions from Levermann et al. (2020) to construct a two-way LRF framework that encompasses the stratification-induced feedback. We use this framework to compute projections of the Antarctic contribution to 21st century sea-level rise. In addition, we re-evaluate the calibration of van der Linden et al. (2023) and the relation between basal melt and ocean temperatures.

This paper is structured as follows. In Section 2 we present the EC-Earth3 model and its treatment of the Antarctic freshwater balance, the experiments used to quantify the meltwater–warming feedback, as well as the linear response function framework. In Section 3 we present the results; here we quantify the feedback between meltwater and ice mass loss and its implications for sea-level projections. In Section 4 we discuss the limitations and implications of the results. Finally, in Section 5 we summarise our findings and conclude with recommendations.

## 2 Methods

In this Section, we describe the relevant specifics of the Earth System Model EC-Earth3 which we use to simulate the impact of meltwater release on subsurface ocean temperatures around Antarctica. Next, we describe the simulations we perform using this model. We continue to describe the sea-level computations using the response function framework. Finally, we re-evaluate the calibration of the basal melt parameter in the temperature-melt relationship.

### 2.1 Model and experiments

#### 2.1.1 EC-Earth3

In this study, we use the ESM EC-Earth3 (based on revision 8153). For all intents and purposes, this model version is identical to the model version used for the CMIP6 experiments (Döscher et al., 2022). The ocean model is NEMO3.6 (Madec et al., 2019), configured on the ORCA1L75 grid, which is a tripolar $1°$ grid with 75 vertical layers. The sea-ice component is LIM3 (Vancoppenolle et al., 2009). This model version is known to exhibit a warm bias in the Southern Ocean, though an intercomparison study showed the bias to be intermediate among CMIP6 models (Heuzé, 2021). In all experiments, the model is run under the general pre-industrial control forcing, with the exception of the freshwater forcing around the Antarctic continent.

### 2.1.2 Original freshwater balance

Like most ESMs, the CMIP6 version of EC-Earth3 lacks a dynamic representation of ice sheets. The model therefore contains no dynamical description of the transformation of accumulating snow into ice, the ice flow towards the coast, and the basal melt and calving of ice leading to a meltwater flux into the ocean surface and subsurface. Rather, this process is represented as an immediate redirection of excess snowfall toward the surface ocean (Fig. 1a) by the runoff mapper (Döscher et al., 2022). Excess snowfall is defined as the amount of snowfall that would increase the snow layer above a given threshold (10,000

95  kg/m$^2$). At each time step, all excess snowfall over the whole Antarctic ice sheet is redistributed homogeneously over a wide band across the Southern Ocean. The heat required to melt this snow is extracted from the surface ocean.

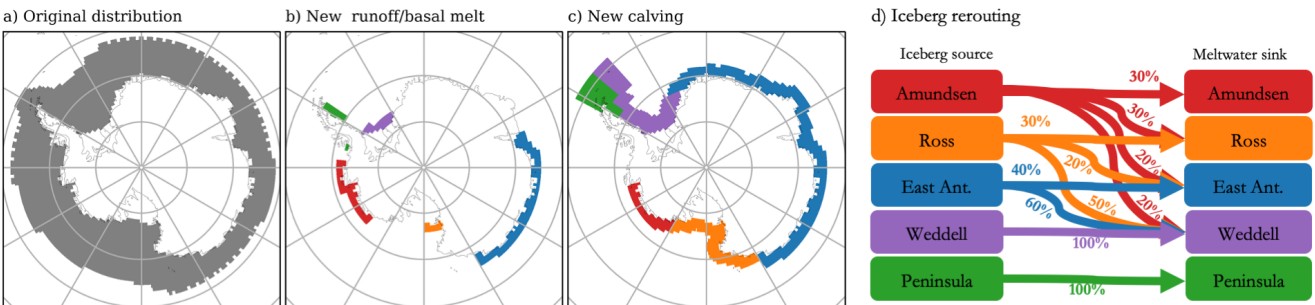

**Figure 1.** Meltwater masks in EC-Earth. a) The original masks applied to all meltwater fluxes from the Antarctic continent, over which meltwater is distributed uniformly at the sea surface. b) New runoff mask, used to distribute basal melt fluxes. These fluxes are distributed locally down to 200 m depth. c) The new calving mask, composed of five regions. Iceberg drift is accounted for by proportionally distributing iceberg source fluxes over downstream meltwater sink regions as expressed in panel d).

The Antarctic snow balance is completed by a direct sublimation of snow and surface snow melt. This melt is distributed over the same surface ocean region, the same way as continental rainfall is rerouted as runoff toward the coastal ocean. Altogether, a balance exists between net snowfall over the Antarctic continent (snowfall−sublimation) and total meltwater input into the

ocean.

### 2.1.3 Modified freshwater balance

As a new control experiment (CTRL), we aim for a more realistic representation of meltwater release due to iceberg calving and ice shelf basal melt. The meltwater fluxes from both calving and basal melt are distributed over five separate regions; these are a modification of the ocean sectors defined by Levermann et al. (2020), as shown in Fig. 1b,c.

The total ice mass loss (basal melt + calving) is chosen to be constant in time, equal to the 450-year average pre-industrial control freshwater balance of the CMIP6 version of EC-Earth3, namely 3300 Gt/yr (1 Gt/yr = 0.0317 mSv). In the CTRL experiment, the internally computed runoff is removed, and replaced by a prescribed flux of 3300 Gt/yr ($\approx$ 0.10 Sv). Our new control run therefore only varies from CMIP6 in the spatial distribution of freshwater around the Antarctic continent and its

interannual variability, but not in its total average magnitude. The distribution of this total meltwater flux between basal melt
and calving, and between the five regions, is fixed using the observed mass loss by Rignot et al. (2013) and is given in Table 1.

**Table 1.** Region-specific parameters. Calving and basal melt are the fractions of the total meltwater flux from Rignot et al. (2013). The ice
shelf depth is the range over which temperature anomalies are diagnosed. These are 100m depth ranges centered around the central values
from Levermann et al. (2014).

| Region | Calving | Basal melt | Sum | Ice shelf depth |
|---|---|---|---|---|
| East Antarctic | 15.5% | 15.5% | 31% | 319–419 m |
| Ross | 6% | 3% | 9% | 262–362 m |
| Amundsen | 9% | 23% | 32% | 255–355 m |
| Weddell | 12% | 7% | 19% | 370–470 m |
| Peninsula | 2.5% | 6.5% | 9% | 370–470 m |
| Total | 45% | 55% | 100% | |

The meltwater distribution from basal melt accounts for the subsurface freshwater input. The basal melt flux is applied as
a subsurface river runoff, distributed down to a depth of 200 m. Horizontally, the freshwater forcing from basal melt from
each of the five regions is concentrated near presently fast-melting ice shelves as identified from Rignot et al. (2013) (Fig. 1b).
The offshore extent of the freshwater forcing is 250 km, comparable to the model distribution of river runoff along the global
coastline.

The distribution of meltwater due to calving is slightly more complex, as icebergs drift, and meltwater release occurs along
its trajectory. The trajectories from icebergs from different source regions overlap, and icebergs from most source regions end
up in the iceberg alley in the western Weddell Sea. To account for this, we create five sink regions (Fig. 1c), which are a
simplified representation of the complex freshwater release from icebergs (e.g., Merino et al., 2016). For each source region, a
distribution of meltwater release between these sink regions is prescribed. Note that this distribution should be interpreted as a
rough approximation, yet a more realistic one than the assumption of spatially homogeneous iceberg melt without taking into
account iceberg drift.

We will evaluate the modified freshwater balance by comparing the ocean temperatures in the CTRL simulation to estimates
from reanalysis data sets. Here, we follow the protocol of van der Linden et al. (2023) and take the average of four products:
GLORYS2V4 (by Mercator Ocean, France), ORAS5 (by ECMWF), FOAM/GloSea5 (by Met Office UK), and C-GLORS05
(CMCC, Italy). Note, however, that due to limitation of observations, each of these products may contain significant biases in
Southern Ocean temperatures.

### 2.1.4 Perturbation experiments

The above-described simulation was run for 100 years (spin-up) and a subsequent 150 years (control, CTRL), starting from
the CMIP6 pre-industrial control simulation. Based on diagnosed subsurface ocean temperatures (Sect. 3.1, Fig. 3), the system
reaches a quasi-equilibrium after the spin-up and is in a relatively stable state throughout the control period.

From this new spin-up at year 100, a set of seven experiments is performed with enhanced freshwater input, for a period of 150 years (see Table 2). As meltwater release varies strongly between the different regions, and regional ocean temperatures may respond differently to meltwater release from different regions, we first design five regional perturbation experiments.

These are based on a step-function increase in ice mass loss of 400 Gt/yr ($\approx$ 0.012 Sv) from either one of the five source regions. The perturbation is divided into basal melt and calving following Table 1 in order to maintain the relative distribution for that specific source region. For example, for the perturbation experiment with increased ice mass loss from East Antarctica (EAIS), the observed distribution between basal melt and calving (both 50%, see Table 1) is maintained by adding 200 Gt/yr ($\approx$ 0.006 Sv) as a basal melt flux, and 200 Gt/yr as a calving flux. Of this calving flux, 60% is exported to the Weddell Sea

(Fig 1d) where it is released as meltwater. These experiments are referred to as EAIS, ROSS, AMUN, WEDD, and PENS, based on the source region of the meltwater perturbation.

In the last two experiments, a step-wise increase in ice mass loss is added simultaneously from all five source regions. Again, the relative distribution between calving and basal melt for each region is fixed to the observed distribution in Table 1. One experiment is performed with an increased meltwater flux of 200 Gt/yr from each region (ALL1, 1000 Gt/yr ($\approx$ 0.032 Sv) in

total). The last experiment contains 400 Gt/yr from each region (ALL2, 2000 Gt/yr ($\approx$ 0.063 Sv) in total). The values of these perturbations are chosen such that the maximum total freshwater increase (2000 Gt/yr) is comparable to the total observed surface mass balance of the Antarctic ice sheet.

**Table 2.** List of experiments.

| Name | Experiment | Model Years |
|------|-----------|-------------|
| piControl | CMIP6 pre-industrial control initial state | $-450$–0 |
| spin-up and CTRL | more realistic (fixed) spatial meltwater distribution | 0–250 |
| EAIS/ROSS/AMUN/WEDD/PENS | 400 Gt/yr from either of the 5 source regions | 100–250 |
| ALL1, ALL2 | 200/400 Gt/yr from each of the 5 source regions | 100–250 |

## 2.2 Sea-level computation

The meltwater-warming feedback is quantified through the computation of the Antarctic sea-level contribution using two

methods: with and without the meltwater-warming feedback (Fig. 2). The computation without feedback can be interpreted as a reduced-physics approximation to stand-alone ice sheet model simulations, whereas the computation with feedback acts as a reduced-physics approximation to coupled ice–ocean simulations. In both cases, ocean warming as simulated by CMIP6 ESMs is translated into an increase in ice shelf basal melt and ultimately to a sea-level contribution.

### 2.2.1 Ocean warming input

The ocean warming input is taken from the same ensemble of 14 ESMs as van der Linden et al. (2023). For each ESM we use the historical simulation starting in 1850, and three future SSP scenarios (SSP1-2.6, SSP2-4.5, and SSP5-8.5) until 2100. From

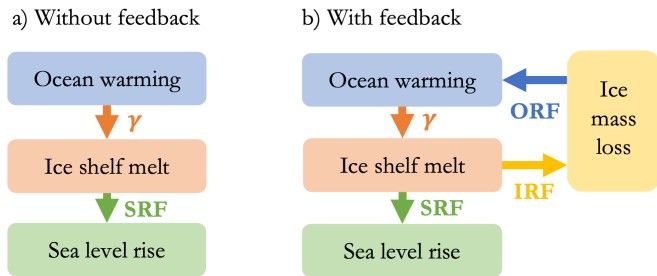

**Figure 2.** Two methods to compute the Antarctic sea-level contribution from ocean warming. a) Method of Levermann et al. (2020) without feedbacks. b) Expanded method accounting for the potential feedback between mass loss and ocean warming. SRF: Sea-level Response Function. IRF: Ice mass Response Function. ORF: Ocean Response Function.

all simulations, the linear trend from the full length of the piControl simulation is removed, in order to remove any potential model drift.

From the 3D ocean fields, the subsurface ocean temperature change is averaged over a set of 5 ocean regions. Horizontally, these ocean regions are identical to those in Levermann et al. (2020); vertically, we take the depth range over which the ice shelf base is assumed to be in contact with the ocean to be smaller, namely 100 m, as expressed in Table 1. A sensitivity analysis to the horizontal extent, including only the regions south of the 1000 m isobath, is described in App. D.

The change in the subsurface ocean temperature $T_j$ in ocean region $j$ is translated into a change in ice shelf basal melt $\dot{m}_i$ of the associated ice sheet region $i$. Here, we use a quadratic relation between basal melt and thermal forcing:

$$\dot{m}_i = \gamma(T_j - T_f)^2; j = i \tag{1}$$

Here, $\dot{m}_i$ is the ice shelf basal melt rate in m yr$^{-1}$. Such a quadratic relation was previously proposed for ice sheet modelling (Favier et al., 2014; Jourdain et al., 2020) and is supported by observations (Jenkins et al., 2018). $T_f$ is the freezing temperature; for simplicity, we take a constant value of -1.7°C throughout which approximates the surface freezing temperature. Note that lower freezing temperatures are more appropriate at the depth ranges studied here. We have not assessed the uncertainty related to this idealised freezing temperature. However, we consider this uncertainty to be minimal in comparison to uncertainties from other methodological assumptions, such as the division of ocean regions into five basins, the application of uniform basal melt rates to five ice sheet sectors, and the quadratic relation between basal melt and thermal forcing itself. A refinement of these latter idealisations is likely to provide a bigger advantage than a refinement of the freezing temperature. The parameter $\gamma$, expressed in m yr$^{-1}$ °C$^{-2}$, determines the basal melt sensitivity and is a key tuning parameter. We apply two ways to determine $\gamma$, one fixed and one calibrated, as described in Sec. 2.3.

### 2.2.2 Without feedback

The method to compute the Antarctic sea-level contribution without the meltwater-warming feedback (Fig. 2a) is similar to that by Levermann et al. (2020) and van der Linden et al. (2023). In this method, ice shelf basal melt is translated directly to a sea-level rise contribution through the Sea Level Response Function (SRF).

The SRF is based on the LARMIP-2 experiments where a set of dynamical ice sheet models was exposed to increased ice shelf basal melt. The SRFs were computed and published by Levermann et al. (2020), based on step-function increases in ice shelf basal melt in five regions. The SRF, denoted by $R_i^S$, is computed following:

$$R_i^S = \frac{1}{\Delta \dot{m}_i} \frac{dS_i}{dt} \tag{2}$$

Here, $\Delta \dot{m}_i$ is the applied perturbation in the basal melt rate, in this case 8 m yr$^{-1}$. $S_i$ is the diagnosed change in the volume
above flotation in metres from ice sheet region $i \in \{1, 2, 3, 4, 5\}$. Note that $R_i^S$ is unitless.

Based on a time series of anomalous basal melt rates $\dot{m}_i$, as derived from simulations of ocean warming (Sec. 2.2.1), a corresponding sea-level contribution from region $i$ can then be computed following:

$$S_i(t) = \int_0^t \dot{m}_i(t') R_i^S(t - t') dt' \tag{3}$$

The sea-level contributions from the individual regions can be added to a total contribution $S(t)$.

### 190   2.2.3 With feedback

For the computation of the Antarctic sea-level contribution with feedback (Fig. 2b), novel to this study, we introduce two additional response functions. The Ice Respone Function (IRF) is the total ice loss, i.e., meltwater release, in response to increased ice shelf melting. The Ocean Response Function (ORF) quantifies the subsurface ocean warming in response to increased meltwater release.

Equivalent to the SRF, the IRF, denoted by $R^I$, is defined as follows:

$$R_i^I = \frac{1}{\Delta \dot{m}_i} \frac{dI_i}{dt} \tag{4}$$

Here, $I_i$ is the cumulative ice mass loss from ice sheet region $i$ in Gt. $R^I$ has units of Gt/m. These IRFs were taken from the same simulations as the SRFs by Levermann et al. (2020).

Similar to the SRF, the IRF can be combined with a time series of anomalous basal melt rate to give a total ice mass loss
from each region.

$$I_i(t) = \int_0^t m_i(t') R_i^I(t - t') dt' \tag{5}$$

The ice mass loss from the individual regions can be added to a total meltwater release $I(t)$.

The ORFs in this study are derived from the perturbation experiments with EC-Earth3, described in Sec. 2.1.4. The ORF expresses the response of the subsurface ocean temperature $\Delta T_j$ in ocean region $j$ to the anomalous ice mass loss rate $\frac{dI}{dt}$, henceforth denoted as $\dot{I}$. The depth ranges of the five regions are specified in Table 1. The ocean temperature response can be expressed based on the regional ice mass loss rate $\dot{I}_i$ or the total ice mass loss rate $\dot{I}$. As we show in Appendix A, the source region of freshwater input has a limited impact on the resultant temperature response. Hence, we here adopt the simpler approach and express the ORF in terms of the total ice mass loss rate $\dot{I}$.

$$R_j^O = \frac{1}{\Delta \dot{I}} \frac{d\Delta T_j}{dt} \tag{6}$$

Here, $\Delta \dot{I}$ is the applied perturbation in the total ice mass loss rate. $\Delta T_j$ is the subsurface ocean temperature response in region $j$. The ORF $R^O$ has units °C / Gt.

For each ocean region $j$, the temperature response to the total anomalous ice mass loss $I$ from all regions can be found through:

$$\Delta T_j(t) = \int_0^t \dot{I}(t') R_j^O(t-t') dt' \tag{7}$$

The ocean temperature anomaly contains a strong internal variability that is unrelated to the applied perturbation. To prevent the erroneous attribution of this variability to the response function, we follow Lambert et al. (2019) and apply a linear fit to the CTRL run and an exponential fit to the ORF $R^O$.

To compute the sea-level contribution with feedback (Fig. 2b), the increased ice shelf melting is translated into an ice mass loss $I$. Next, the ice mass loss rate $\dot{I}$ is used as input for the ORF to determine a time series for the ocean temperature anomaly $\Delta T_j$ using Eq. 7. This meltwater-induced temperature anomaly is then added to the initial ESM-based temperature time series.

The above application of $\gamma$, IRF, and ORF is then repeated iteratively, until the time series converge, producing a set of time series in which the simulated ice mass loss $I$ is consistent with the temperature $T_j$, and the temperature anomaly $\Delta T_j$ – relative to the ESM-based temperature – is consistent with the ice mass loss rate $\dot{I}$. Finally, from the revised temperature time series, the SRF is applied in order to compute the cumulative sea-level contribution $S$.

LARMIP-2 was based on a set of 16 ice sheet models providing SRFs. For this study, we received IRFs from a total of 8 modelling groups. For consistency, we use this same ensemble of 8 models in the computations with and without feedback. Note that all ORFs are derived from a single ESM, EC-Earth3. In combination with ocean warming from 14 ESMs, this gives us a total ensemble size of 112 ESM-ISM model pairs from which we derive a probability distribution.

## 2.3 Basal melt parameter $\gamma$

The parameter $\gamma$ (Eq. 1), relating ice shelf basal melt rates to ocean temperatures, is a key source of uncertainty in sea-level rise projections (Seroussi et al., 2020). Here, we apply two ways to determine $\gamma$: one fixed $\gamma_f$ and one calibrated $\gamma_c$.

### 2.3.1 Fixed $\gamma_f$

The fixed $\gamma_f$ is applied to sea-level computations with and without feedback based on all ESM-ISM model pairs. This allows for a direct quantification of the magnitude of the meltwater–warming feedback. As $\gamma_f$, we use the median value from Jourdain et al. (2020), calibrated on observed mean basal melt rates, averaged over the Antarctic ice shelves. This median value, corresponding to $\gamma_f = 2.57$ m yr$^{-1}$ °C$^{-2}$, is used in the nonlocal parameterisation in ISMIP6 (Nowicki et al., 2020). Note that this parameterisation is in fact the local-nonlocal parameterisation proposed by Favier et al. (2019), while the present study uses a fully nonlocal parameterisation, based on large-scale ocean temperatures only.

### 2.3.2 Calibrated $\gamma_c$

The calibration of $\gamma$ provides a second way to quantify the meltwater–warming feedback. By determining $\gamma_c$ in the sea-level computation with and without feedback, the difference in $\gamma_c$ is a metric for the magnitude of the feedback. To calibrate $\gamma_c$, we quantify the historical ice mass loss over the period 1979–2017 based on the modelled ocean warming (including the meltwater–warming feedback) over the period 1871–2020 and adjust parameter $\gamma_c$ to match the observed historical ice mass loss.

The calibrated $\gamma_c$ is determined separately for each ESM-ISM model pair. This method is largely based on van der Linden et al. (2023) and calibrates $\gamma_c$ on the observed Antarctic sea-level contribution over the period 1979–2017 (Rignot et al., 2019). To determine $\gamma_c$, we first apply an initial guess for $\gamma$ and convert ESM-derived ocean temperatures between 1871–2020 to sea-level rise. From this, the cumulative Antarctic sea-level contribution between 1979 and 2017 is diagnosed for each ISM. Next, $\gamma$ is adjusted until the value $\gamma_c$ is found which leads to the observed cumulative sea-level contribution of 1.32 cm. Note that the uncertainty in this value is negligible in comparison to uncertainties stemming from simulated ocean warming, and any methodological choices in this study. This method produces individual values of $\gamma_c$ for each ESM-ISM model pair, and different values when the feedback is included or not. Note that this calibration method is similar but not identical to that of van der Linden et al. (2023), as their calibration is based on the root mean squared error of the full time series of cumulative sea-level rise while we only use the total rise over the full period to calibrate the basal melt parameter.

As described by van der Linden et al. (2023), the calibration results in a negative value of $\gamma_c$ for some model pairs. This occurs primarily when a given ESM simulates a net cooling over the historical period in some regions. Although a net historical cooling may very well be realistic, the resultant negative parameter $\gamma_c$ is unrealistic. We therefore deem these model pairs invalid for our calibration method; these are therefore omitted from our subset. After this selection, an ensemble size of 85 model pairs is retained for further analysis.

## 3 Results

In this Section, we present the results of our simulations and the derived computations of the Antarctic sea-level contribution based on ocean warming. In Sec. 3.1, we analyse the impact of the revised meltwater distribution on subsurface ocean temper-

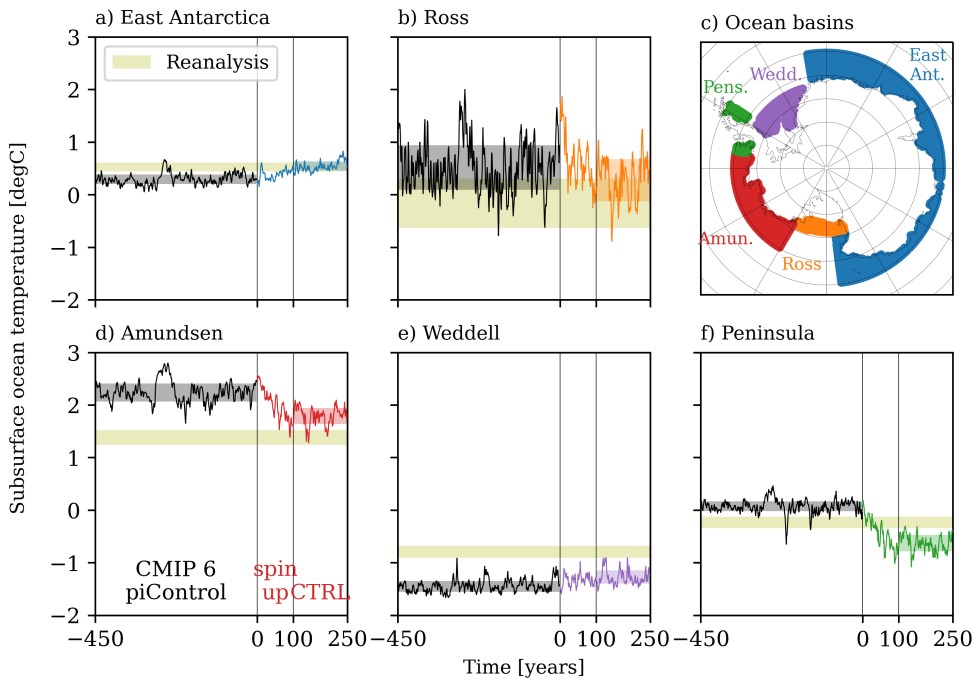

**Figure 3.** Control time series of subsurface ocean temperatures from EC-Earth3. Black lines in (a,b,d,e,f) are the last 150 years of the CMIP 6 pre-industrial control runs. Coloured lines indicate the simulated spinup (years 0 to 100) and CTRL (years 100 to 250), with an equal yet redistributed meltwater flux. Black and coloured shading indicate the 17th to 83rd percentiles. Olive shading indicates the 17th to 83rd percentile range of four combined reanalysis data sets (GLORYS2V4, ORAS5, FOAM/GloSea5, and C-GLORS05, van der Linden et al., 2023) for the respective regions. c) Corresponding ocean sectors largely following Levermann et al. (2020).

atures. In Sec. 3.2, we describe the impact of the meltwater perturbations on subsurface ocean temperatures and construct the ORFs. Finally, in Sec. 3.3, we quantify the impact of the meltwater–warming feedback on the Antarctic sea-level contribution 265 and the impact of this feedback on the calibrated basal melt parameter $\gamma_c$.

## 3.1 Evaluation of the modified freshwater balance

In our experiments, the meltwater distribution from calving and basal melt is distributed more realistically than in the EC-Earth3 simulations for CMIP6. The meltwater distribution is enhanced by dividing it into separate regions of basal melt and calving, associated with ocean forcing in five ocean sectors. The meltwater masks are adapted to better represent observed 270 freshwater fluxes due to iceberg melt and basal melt from ice shelves. Iceberg drift is accounted for in the translation from source to sink regions. The fractionation between calving and basal melt and between the five source regions is based on observations. Finally, the meltwater flux due to basal melt is applied at depth, rather than at the surface.

This modified meltwater treatment may be expected to affect the state of the Southern Ocean. As we are particularly interested in the subsurface ocean temperatures in the five sectors that govern the ice shelf melting, we diagnose these temperatures in Fig. 3. The CMIP6 pre-industrial control simulation shows substantial biases in subsurface temperatures relative to reanalysis-based estimates. The Amundsen, Ross, and Peninsula regions are relatively warm, whereas the East Antarctic and Weddell regions are relatively cold. Note that in practice, the Peninsula region is a weighted average between the Amundsen and Weddell regions, and any biases can be understood from the biases in these respective regions. As we will show in Sec. 3.3.1, the contribution of the Peninsula to the meltwater–warming feedback and sea-level contribution is minimal, hence, we do not investigate the individual responses in the eastern and western Peninsula separately.

After spinup, we observe that the warm biases in the Amundsen and Ross regions decrease, most substantially in the former. The slight cold bias in the East Antarctic region disappears completely, and that in the Weddell region decreases slightly. Due to the cooling of the Amundsen region, the warm bias in the Peninsula region changes into a cold bias of comparable magnitude, dominated by the cold bias on the Weddell side. Besides the mean states, the variability in all regions agrees well with the variability from the reanalysis, as represented by the width of the shaded bands.

In the East Antarctic, Ross, and Weddell regions, increased stratification leads to a cooling in the top few hundred meters, and a warming below (Fig. 4). The depth range used for the basal melt computation in the East Antarctic and Weddell regions crosses the warming signal, whereas the shallower depth range for Ross crosses the cooling signal. For the Amundsen region and the western Peninsula, reduced stratification due to dominant subsurface meltwater forcing induces a warming in the top 100 m and a cooling below, which is reflected in the net cooling signal in Fig. 3. These differences in stratification explain the temperature adjustment in the control state, which generally lies closer to the reanalysis-based estimates in the five ocean regions.

The reduced biases in nearly all subsurface temperatures strengthen our confidence that our meltwater forcing is relatively realistic, at least compared to the CMIP6 version of EC-Earth3. These reduced biases may result from the more localised meltwater distribution close to the coastline, or the revised distribution of meltwater distribution in terms of basal melting and calving from the five separate regions. However, more analysis is required to assess whether these reduced biases in fact reflect a model improvement overall. This analysis goes beyond the scope of this study, though the improved match between simulated ocean temperatures and those from the various reanalysis products provides a robust starting point for the perturbation experiments.

## 3.2 Meltwater perturbation impact on ocean

### 3.2.1 Ocean Response Functions

The response of the subsurface ocean temperatures in the five regions to enhanced meltwater forcing is shown in Fig. 5. In the left column (400 Gt/yr), the five regional perturbation experiments EAIS, WEDD, etc. are displayed as grey lines. These five regional experiments are shown in more detail in Fig. A1. First and foremost, we find that an increase in meltwater release unequivocally leads to a subsurface ocean warming. This indicates that according to EC-Earth3, the meltwater-warming

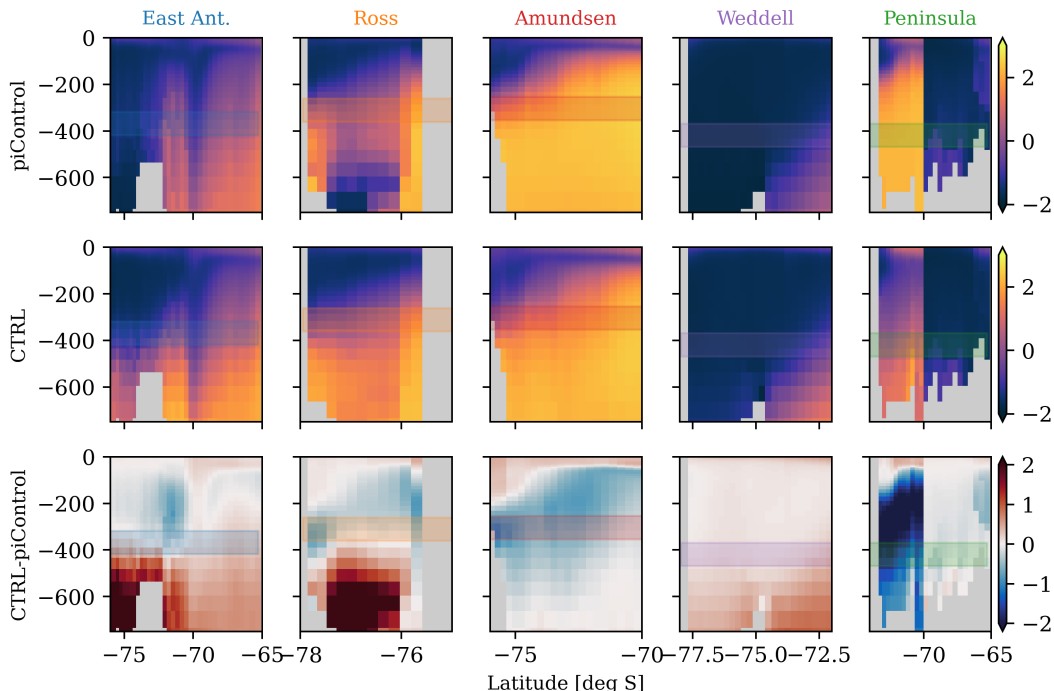

**Figure 4.** Zonal mean potential temperature in EC-Earth3 (shading in degrees Celcius) with latitude on the x-axis and depth on the y-axis. Columns denote the five ocean sectors (see Fig. 3c). The top row shows the CMIP6 piControl average over 450 years. The middle row shows the new CTRL, averaged from year 100 to year 250. The bottom row shows the difference of the new CTRL with respect to the CMIP6 piControl. Shaded bands denote the depth range at which temperature anomalies are taken to parameterise basal melt rates. The latitudinal extent is defined by the ocean sectors. Note that the Peninsula cross-sections are split between the western Peninsula (south of 70S) and the eastern Peninsula (north of 70S).

feedback is a positive one along the whole continent. We note that the magnitude of this warming response is conditional on the choice of ESM. While ESMs in general simulate a subsurface warming due to enhanced meltwater release, the magnitude and extent toward the coast of this warming differs strongly between ESMs (Chen et al., 2023).

A second observation in Fig. A1 is that the magnitude of the warming response in EC-Earth3 in each region appears to be independent of the source region of the meltwater perturbation. The time scale of the adjustment does appear to vary, yet with the substantial multidecadal variability in the model, it is difficult to detect a significant difference in adjustment time scales. Hence, as mentioned in Sec. 2.2.3, we proceed with ORFs based on the total meltwater release $I(t)$, which is represented by the coloured lines in Fig. 5.

For the average 400 Gt/yr experiment, a large range in response magnitudes is found. The ocean warming in the Amundsen Sea is minimal, as the stratification there is already strong (Fig. 4) and additional meltwater release has a weak potential to

strengthen this stratification. Notably, other ESMs even show a regional cooling of the subsurface Amundsen Sea in response to a meltwater flux increase (Beadling et al., 2022; Thomas et al., 2023). Hence, the sign of the Amundsen Sea response is ambiguous amongst different ESMs.

In contrast, the ocean warming response in EC-Earth3 in the Ross Sea is an order of magnitude larger. This region appears to be in a sensitive control-state and the same meltwater perturbation has a significantly larger potential to strengthen the stratification and increase the subsurface ocean temperature. Hence, we may expect the meltwater–warming feedback to have a larger impact on mass loss from the Ross region than on the Amundsen region.

### 3.2.2 Linearity check

In order to test the validity of the ORF, we must determine whether the response to perturbations is linear. This linearity is tested by comparing the average 400 Gt/yr experiment to the two experiments with a total meltwater input of 1000 and 2000 Gt/yr (Fig. 5).

For the three perturbation magnitudes, the equilibrium response is diagnosed by averaging the temperature anomaly over the years 75–150 (horizontal lines). In the rightmost column, these equilibrium responses are displayed by the solid dots, as a function of the total meltwater perturbation. We observe a saturation effect for large perturbations, as reported by Schloesser et al. (2019) . For each ocean region, the temperature response to 2000 Gt/yr meltwater input is substantially weaker than five times the temperature response to 400 Gt/yr meltwater input.

Applying the ORFs directly as linear response functions (denoted by the dotted lines in Fig. 5) would lead to an overestimation of the response to large meltwater variations. To prevent this, we apply an upper limit $\dot{I_m}$ to $\dot{I}$. This correction is illustrated by the solid lines and the value of $\dot{I_m}$ is determined to best fit the equilibrium response to the different perturbation magnitudes. Note that we have also tested a nonlinear fit, which is included in the analysis code (Lambert, 2025). However, the impact was minimal, so we opted for the simpler and more transparent option of a linear fit with upper limit.

Note that the saturation effect found here partly compensates for the non-linearity in the basal melt parameterisation (Eq. 1). This quadratic relation between temperature and ice shelf melting prescribes a relatively large sensitivity in warm regions like the Amundsen Sea. In contrast, the saturation effect illustrates that regions with an already strong meltwater input and stratification, like the Amundsen Sea, are less sensitive to a further increase in meltwater input. The meltwater–warming feedback in the Amundsen Sea is therefore governed by a moderate increase in ocean temperature, but a relatively strong increase in meltwater release. In contrast, the feedback in the cold Weddell Sea is governed by a relatively strong increase in ocean temperature, but a more moderate increase in meltwater release.

### 3.3 Feedback quantification

To quantify the meltwater–warming feedback, we take two approaches: first, we compute the sea-level projection with and without feedback with a fixed $\gamma_f$ (Sec. 3.3.1).

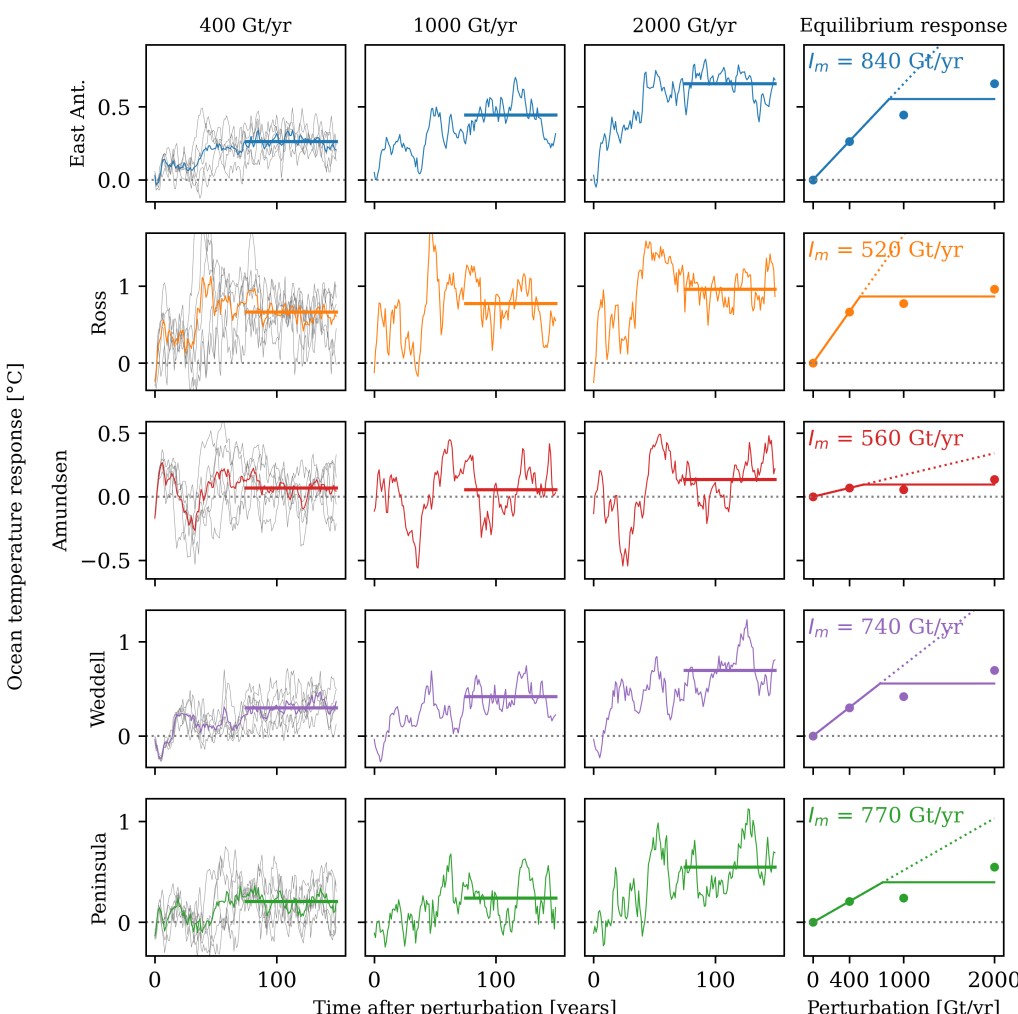

**Figure 5.** Ocean Response Functions (ORFs) and linearity check based on EC-Earth3. Each column denotes equivalent ORFs for the five ocean regions. Column 1 (400 Gt/yr) shows the five individual perturbation experiments (grey) and the average (colors) as displayed in Fig. A1. Columns 2 and 3 are the results of the experiments with a total of 1000 and 2000 Gt/yr perturbation respectively. The last column expresses the equilibrium response (last 75 years) of the first columns, as a function of the perturbation magnitude. The dashed line represents the linearity assumption, the solid line is the corrected relation. Note the different scale on the y-axis for the different regions.

Second, we calibrate $\gamma_c$ with and without feedback to reproduce the historical sea-level contribution from Antarctica (Sec. 3.3.2). This method of quantification can provide insight into what basal melt parameters should be used in stand-alone and coupled model setups.

### 3.3.1 Impact on sea level projections

In order to illustrate the impact of the feedback on sea-level projections, we highlight a single representative ensemble member. Fig. 6 displays the time series of ocean temperature, ice mass loss and the Antarctic sea-level contribution based on ocean temperatures from EC-Earth3 under SSP2-4.5 using the IRF and SRF from CISM-NCA, which exhibits an intermediate ice mass loss response to basal melting (see Fig. B1). The ice mass loss and sea-level contribution without feedback are directly derived from the temperature time series through conversion to basal melt using $\gamma_f = 2.57$ m yr$^{-1}$ °C$^{-2}$ (Sec. 2.3.1).

In the inclusion of the feedback (coloured lines), a solution is found in which the ice mass loss is consistent with the base ocean temperature plus the subsurface warming due to additional freshwater. In all regions, the temperature increase and ice mass loss are enhanced due to the positive meltwater–warming feedback. Finally, the sea-level contribution is consistent with both temperature and ice mass loss, and is also enhanced compared to the computation without feedback.

In this representative example, we observe that the increased sea-level contribution is dominated by the positive feedback from the Ross Sea and to a lesser extent the East Antarctic region. The contribution from the Amundsen Sea is limited, as the meltwater–warming feedback is weak in this region, due to the already strong present-day stratification. The contribution of the Peninsula is weak, because of the overall limited ice available, represented by a small IRF. Finally, the contribution of the Weddell Sea is limited despite the relatively strong temperature response, as the quadratic melt parameterisation suppresses the impact of warming on ice shelf basal melt in cold regions.

Both the Ross Sea and the East Antarctic region are warm enough to allow for a significant impact of ocean warming on ice shelf basal melt. In addition, the present-day stratification is relatively weak, allowing for a relatively strong impact of additional meltwater release on intensifying the stratification and warming the sub-surface ocean. Hence, these regions are in the optimal regime to allow for a strong meltwater–warming feedback.

The above procedure is applied to 112 ESM-ISM model pairs and to 3 SSP scenarios (Fig. 7). The median projection with feedback is approximately 80% higher than without the feedback in 2100. This same amplification applies to all SSP scenarios which show a moderate spread in comparison to the estimated uncertainties. The strong impact of the feedback is dominated by the Ross Sea and the East Antarctic regions, as exemplified in Fig. 6. Note that neither projection with or without feedback should be interpreted as a directly relevant sea-level projection, as $\gamma_f$ is not calibrated on historical basal melt or sea-level rise. Rather, these projections illustrate the magnitude and significance of the positive meltwater–warming feedback. Critically, it illustrates the impact of moving from a stand-alone experiment to a coupled ice-ocean experiment without adjusting $\gamma$. To illustrate this, we will quantify the impact of the feedback on the calibration of $\gamma_c$.

### 3.3.2 Impact on basal melt sensitivity

In order to produce more reliable projections of sea-level rise, we calibrate $\gamma_c$ on historical sea-level rise. This method largely follows van der Linden et al. (2023), and calibrates each ESM-ISM model pair individually. In this study, we calibrate $\gamma_c$ on the cumulative historical sea-level rise between 1979 and 2017. This calibration is illustrated in Fig. 8, in which all ensemble

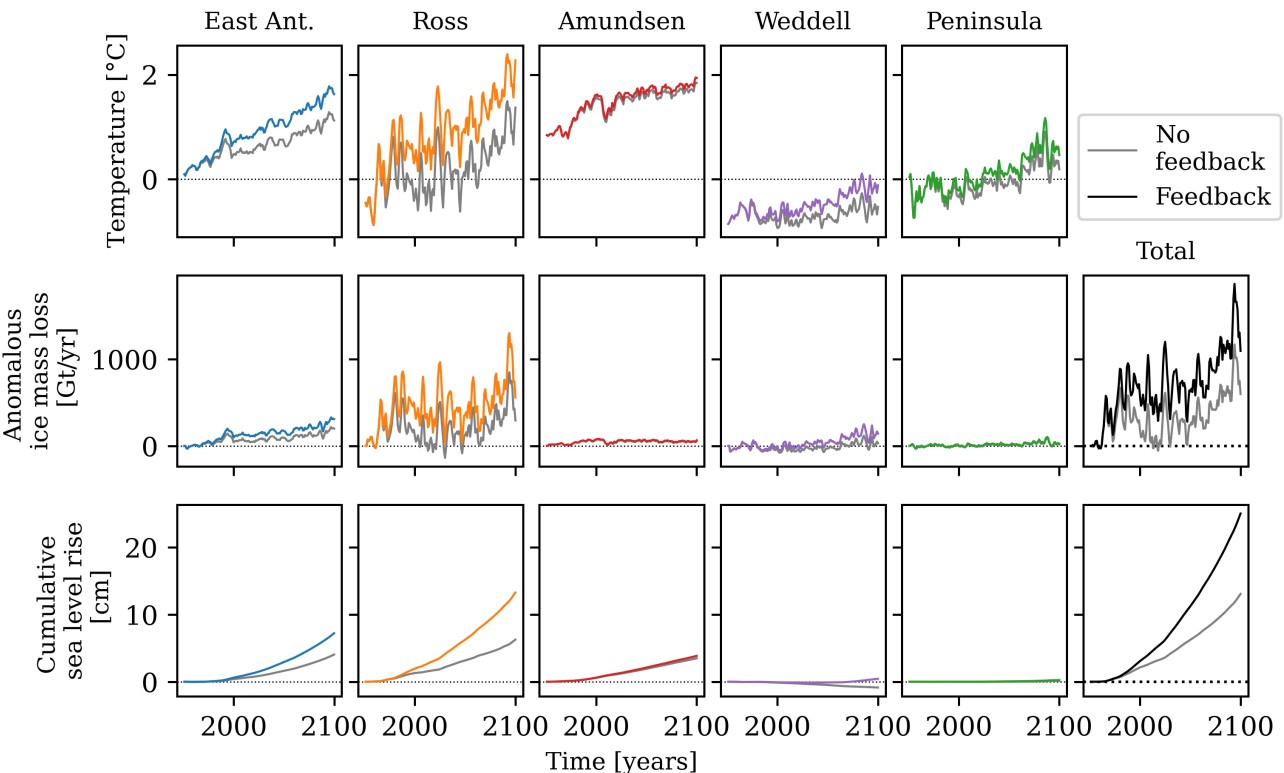

**Figure 6.** Impact of feedback on a single ensemble member based on the temperature projection from EC-Earth3 under SSP2-4.5, IRF and SRF from ice model CISM-NCA and with quadratic basal melt parameterisation. Time series are shown for subsurface temperature, cumulative ice mass loss, and cumulative sea-level rise from five regions. The grey lines indicate the response without feedback. Coloured lines (denoting five regions in Fig. 3c) indicate the response including the feedback.

members with and without feedback are assigned an individual $\gamma_c$ which reproduces the cumulative sea-level contribution of 1.32 cm.

The resulting values of $\gamma_c$ are presented in Fig. 9. The calibrated $\gamma_c$ without feedback is directly comparable to the calibrated
$\gamma$ of van der Linden et al. (2023). The difference between the two is caused by the smaller ISM ensemble size in the present study and the slight difference in the calibration method. Despite this difference, the two $\gamma_c$ distributions are similar, with median values of 2.3 and 2.1 m yr$^{-1}$ °C$^{-2}$. These median values are comparable, though slightly smaller than the applied fixed $\gamma_f$ taken from Jourdain et al. (2020).

In the calibration process, the value of $\gamma_c$ is reduced for each ESM-ISM model pair when the feedback is included, in order
to achieve the same historical sea-level contribution. Hence, the distribution of $\gamma_c$ is reduced by approximately 35%, leading to a median value of 1.7 m yr$^{-1}$ °C$^{-2}$ and a narrower inter-quartile range.

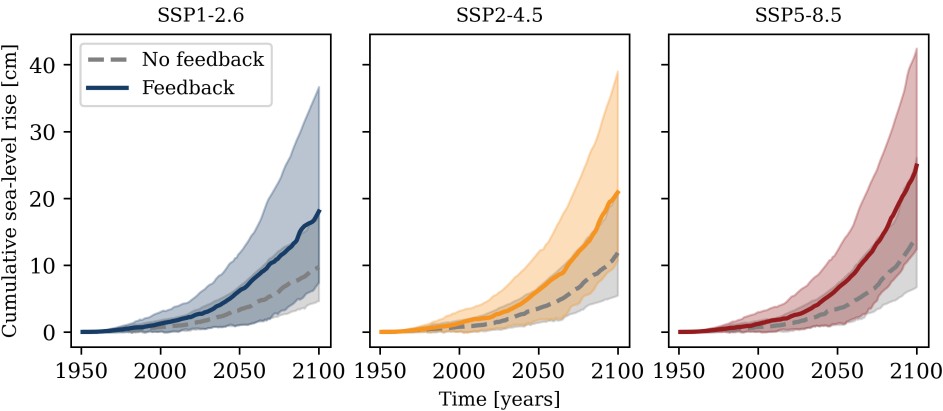

**Figure 7.** Ensemble of sea-level projections with fixed basal melt parameter $\gamma_f$. Projections for three SSP scenarios (columns) are shown. Median (solid line) and 17-83 percentiles (shading) quantify the spread in combinations of 14 Earth System Models and 8 Ice Sheet Models. Grey lines indicate projections without feedback, coloured lines include the feedback.

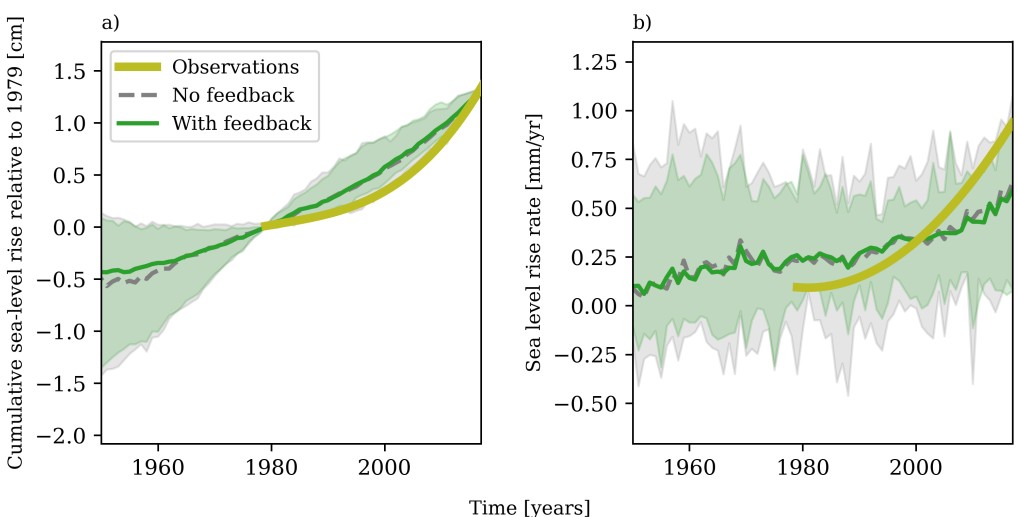

**Figure 8.** Historical sea-level rise computations with calibrated $\gamma_c$. Left: cumulative sea-level rise, right: sea-level rise rate. Observations are derived from Rignot et al. (2019). These values are the sum of all Antarctic regions, and are based on historical ocean temperatures from EC-Earth3.

These newly calibrated $\gamma_c$ values can again be used to compute future sea-level projections under the three SSP scenarios, as shown in Fig. 10. After calibration, the feedback increases the median sea-level projection by approximately 5% in 2100 for all SSP scenarios. This difference is primarily due to the increased sea-level rise rate in the last decades of the 21st century, implying that the impact of the feedback may be larger beyond 2100.

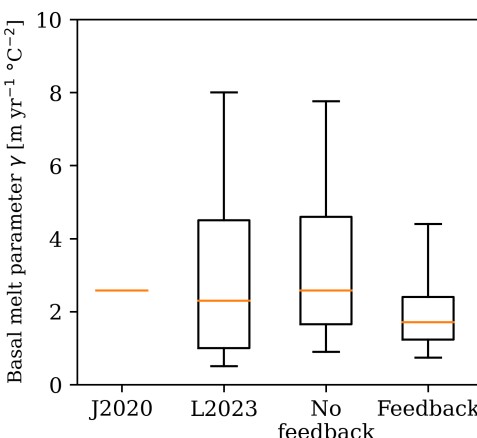

**Figure 9.** Antarctic-wide basal melt parameters $\gamma$. J2020 denotes the median value from Jourdain et al. (2020), used as fixed $\gamma_f$. L2023 denotes van der Linden et al. (2023), used as reference distribution for calibrated $\gamma_c$ without feedback. The latter two distributions are calibrated $\gamma_c$ from this study, with and without feedback. The boxes denote the 25-75th percentiles, the whiskers the 5-95th.

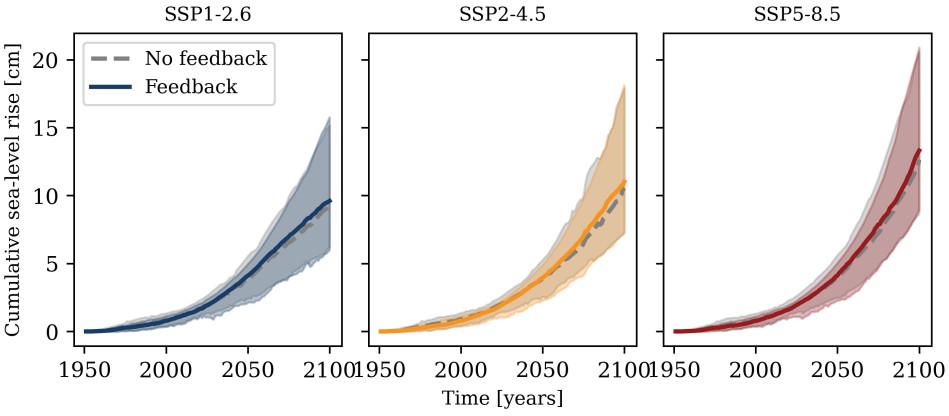

**Figure 10.** Ensemble of sea-level projections with calibrated basal melt parameter $\gamma_c$. Projections for three SSP scenarios (columns) are shown. Median (solid line) and 17-83 percentiles (shading) quantify the spread in combinations of 14 Earth System Models and 8 Ice Sheet Models. Grey lines indicate projections without feedback, identical to those in Fig. 7. Coloured lines include the feedback.

## 4 Discussion

The meltwater forcing in our experiments is highly idealised because it depends on the large regions for which the IRFs and SRFs were computed by Levermann et al. (2020). Rather than explicitly modelling the path of icebergs, the calving flux is

instantaneously distributed in large areas, roughly representing the place where we expect icebergs to melt. We also assume the ratio between basal melt and calving to remain constant in time. These assumptions could have an important influence on our results. However, the relative independence of the ORFs to the source region indicates that these choices have a limited effect on the subsurface warming. Hence, we do not expect a more sophisticated meltwater distribution to qualitatively alter our results and conclusions.

While we take into account the surface cooling of the ocean associated with calving, the cooling associated with basal melt is not taken into account, because for technical simplicity we decided to use the existing runoff mapper in EC-Earth3. This missing negative feedback could lead to an overestimation of our sea-level projections (Pauling et al., 2017) through overestimating ocean warming and thus basal melting. To approximate the magnitude of this negative feedback, a model that resolves the ocean circulation in ice shelf cavities and consequential basal melt is necessary to properly assess the region of heat extraction (e.g., Mathiot et al., 2017). Such advancements in ocean modelling are beyond the scope of this study.

The IRFs and SRFs are forced with a uniform basal melt perturbation across five regions. This basal melt perturbation is based on a simple relation to the thermal forcing, derived from far-field ocean temperatures (Levermann et al., 2020). This basal melt representation can be improved significantly by taking into account a more appropriate extrapolation into ice shelf cavities (Jourdain et al., 2020) and a more sophisticated calculation of basal melting (e.g., Lambert et al., 2023). To incorporate such improvements requires a new suite of model experiments in a similar setting like LARMIP-2 and a new calculation of the IRFs and SRFs.

Our ORFs are based on a single ESM EC-Earth3, which is an inherent limitation of this study. In this ESM, the ocean component contains a horizontal resolution of $1°$ in the zonal direction which is insufficient to resolve the baroclinic Rossby radius of deformation in the Southern Ocean (Hallberg, 2013) and the Antarctic slope current (Thompson et al., 2018). Particularly the response in the Amundsen Sea differs qualitatively between ESMs. Some ESMs show a similar, moderate subsurface warming to EC-Earth3 (Fogwill et al., 2015; Bronselaer et al., 2018), others a negligible response (van Westen and Dijkstra, 2021), whereas several ESMs show a subsurface cooling response in the Amundsen Sea (Moorman et al., 2020; Beadling et al., 2022; Thomas et al., 2023). This inter-model discrepancy indicates that for the Amundsen Sea, the ORF is dependent on the choice of ESM, both in magnitude and sign. However, because of the weak response in EC-Earth3, this region contributes little to the larger-scale meltwater–warming feedback and its contribution to sea-level rise. We therefore expect the Amundsen Sea contribution to the total feedback not to be significantly dependent on the choice of ESM.

In contrast, the meltwater–warming feedback in this study is dominated by the Ross Sea and the East Antarctic region. An inter-model comparison of zonal mean temperature changes within the SOFIA initiative (Swart et al., 2023) showed a qualitative agreement on subsurface warming around Antarctica (Chen et al., 2023). The major disagreement in that intercomparison was the extent to which the warming signal propagates to the continental shelf, as discussed in detail by Beadling et al. (2022). As shown in App. D, including or excluding temperatures offshore from the continental shelf does not qualitatively affect our results. Our quantification of the meltwater–warming feedback based on EC-Earth3 should be interpreted as a first estimate in terms of its potential contribution to sea-level projections. A full probabilistic quantification would benefit from the sampling of ORFs based on a suite of ESMs. For this, the experiments within the SOFIA initiative can be used as a starting point.

Despite the aforementioned caveats and the fact that our results are derived from a single ESM, a number of studies support

our qualitative and quantitative assessment of the meltwater–warming feedback: Observations in the Southern Ocean indicate a surface cooling and a subsurface warming (e.g., Auger et al., 2021), which is generally attributed to increased meltwater forcing (e.g., Schmidt et al., 2023). Quantitatively, the impact of this feedback on the Antarctic sea-level contribution with a fixed $\gamma_f$, approximately +80%, is comparable to the value found in a previous ESM modelling study (+80–100%, Golledge et al., 2019). Bronselaer et al. (2018) found a lower range (9–34%) but with a very large freshwater flux which may have

saturated the ocean response. In addition, the good agreement between calibrated $\gamma_c$ without feedback and the ISMIP 6 values (Jourdain et al., 2020) provides confidence in the robustness of our calibration method. Finally, the simulated median Antarctic sea-level contribution (10–12 cm) is in close agreement with the last IPCC estimates (Fox-Kemper et al., 2021).

Our results can provide modellers with different insights. First, the 80% increase in the 21st century Antarctic sea-level contribution using a fixed $\gamma_f$ illustrates the large potential of the meltwater–warming feedback. Although this number likely

depends on methodological choices including the choice of ESM, our results show that this feedback is non-negligible. This feedback therefore introduces an inconsistency between simulations with stand-alone ISMs and those with coupled ESM-ISMs. We propose that this inconsistency can be partly compensated by using different values for basal melt sensitivities to changes in ocean temperatures. Higher sensitivities in stand-alone ISM simulations can partly compensate for the lack of a meltwater– warming feedback, whereas coupled simulations should use a lower sensitivity to prevent overestimating Antarctica's sea-level

contribution. The calibration method used in this study, leading to a 35% reduction in basal melt parameter $\gamma_c$ is one example to provide consistent historical sea-level rise estimates with and without feedback. The remaining discrepancy due to including or excluding the feedback amounts to a relatively small value of 5% in the sea-level contribution in 2100.

## 5    Conclusions

We have performed an idealised study to quantify the feedback between Antarctic meltwater release and subsurface Southern

Ocean warming. The impact of meltwater release on ocean temperatures was quantified using perturbation experiments with EC-Earth3. These results were combined with response functions of ice mass loss and sea-level rise to compute a probability distribution of the future Antarctic sea-level contribution with and without feedback.

We observe that a more refined distribution of the meltwater flux can significantly alter subsurface ocean temperatures in a control simulation; for EC-Earth3, this causes temperature biases with respect to reanalysis estimates to decrease in

all regions. The perturbation experiments show that, according to the EC-Earth3 model, the meltwater–warming feedback is unambiguously positive, regardless of the meltwater source region. All ocean regions show a subsurface warming response to enhanced meltwater release, with the largest warming in the Ross region and the weakest warming in the Amundsen region.

When a fixed basal melt parameterisation is applied to convert ocean temperatures to ice shelf basal melt, the Antarctic contribution to 21st century sea-level rise increases by 80% when the meltwater–warming feedback is included. The inclusion

of this feedback is an idealised analogy to the transition from stand-alone ice sheet modelling to coupled ice–ocean simulations.

Hence, a similar increase in sea-level contribution may be expected when modelling groups switch from a stand-alone to a coupled setup without adjusting their basal melt parameterisation.

However, we propose that different basal melt parameters should be applied in stand-alone or coupled configurations. One way to adjust the basal melt parameters is to calibrate historical sea-level rise to observations. In our case, this resulted in a basal melt parameter which is 35% lower when the meltwater–warming feedback is included. We consider this an appropriate adjustment for ice sheet modeling studies when switching from a stand-alone to a coupled setup. After the above-described calibration, we find that the feedback enhances the median Antarctic sea-level contribution until 2100 with 5%. This increase is concentrated in the latter half of the century, implying that the meltwater–warming feedback may have a larger impact on sea-level rise beyond 2100.

Our results imply that projections of stand-alone ice sheet modelling ignoring the meltwater–warming feedback may slightly underestimate the 21st century sea-level contribution. A larger implication, though, is that for coupled ESM–ISM modelling. Basal melt parameters are typically calibrated on steady-state basal melt estimates from, e.g., altimetry. This approach, comparable to our method based on the fixed basal melt parameter, leads to a strong discrepancy between stand-alone and coupled modelling. The calibration on integrated mass loss over a given time period allows for a better agreement between the two modelling approaches. We consider this insight particularly relevant for the protocol that is presently in development for the next round of ISMIP (ISMIP7).

## Appendix A:  Individual Ocean Response Functions

As there are five ice sheet regions $i$ and five ocean regions $j$, a total of 25 ORFs are computed.

$$R_{i,j}^O = \frac{1}{\Delta \dot{I}_i} \frac{d\Delta T_j}{dt} \tag{A1}$$

Here, $\Delta \dot{I}_i$ is the applied perturbation in the ice mass loss rate from ice sheet region $i$. In our case, ORFs are based on perturbation experiments of 400 Gt/yr. $\Delta T_j$ is the subsurface ocean temperature response in region $j$. The ORF $R^O$ has units °C / Gt.

For each ocean region $j$, the temperature response to anomalous ice mass loss from all regions $i$ can be found through:

$$\Delta T_j(t) = \sum_i \int_0^t \dot{I}_i(t') R_{i,j}^O(t-t') dt' \tag{A2}$$

The regional temperature response $\Delta T_j$ shows a consistent response, both in magnitude and time scale, which is largely independent on the source region $i$ of the ice mass loss (Fig. A1). Hence, we have chosen to merge these experiments and express the ORFs as a function of the total ice mass loss rate $\dot{I}$. We point out that the response time scale of the Ross Sea appears to vary significantly between the experiments. However, this variation is partly due to a different phasing of interannual

variability. Such interactions between perturbations and natural variability are beyond the scope of this study and are difficult
to express in terms of linear response functions. Hence, we consider this a relevant topic for future research.

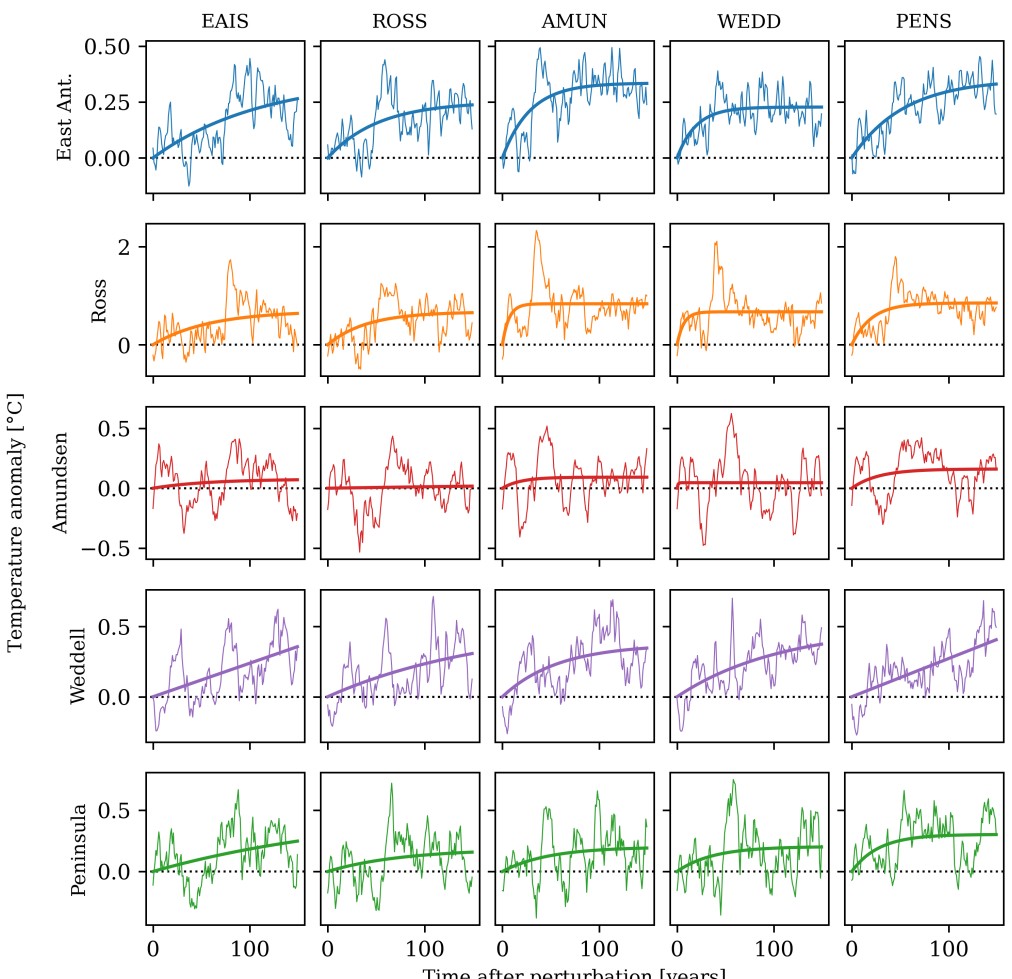

**Figure A1.** Temperature anomalies due to the five perturbation experiments. Each column shows one perturbation experiment, e.g. AMUN denotes the experiment in which 400 Gt/yr enhanced meltwater is released from the Amundsen sector of the ice sheet. Each row represents the temperature response in one of the five ocean basins and an exponential fit. The ORF $R^O_{i,j}$ are computed through time-derivation of these exponential curves. Colours of the temperature response refer to the regions defined in Fig. 3c. Note the different y-scales for the different regions.

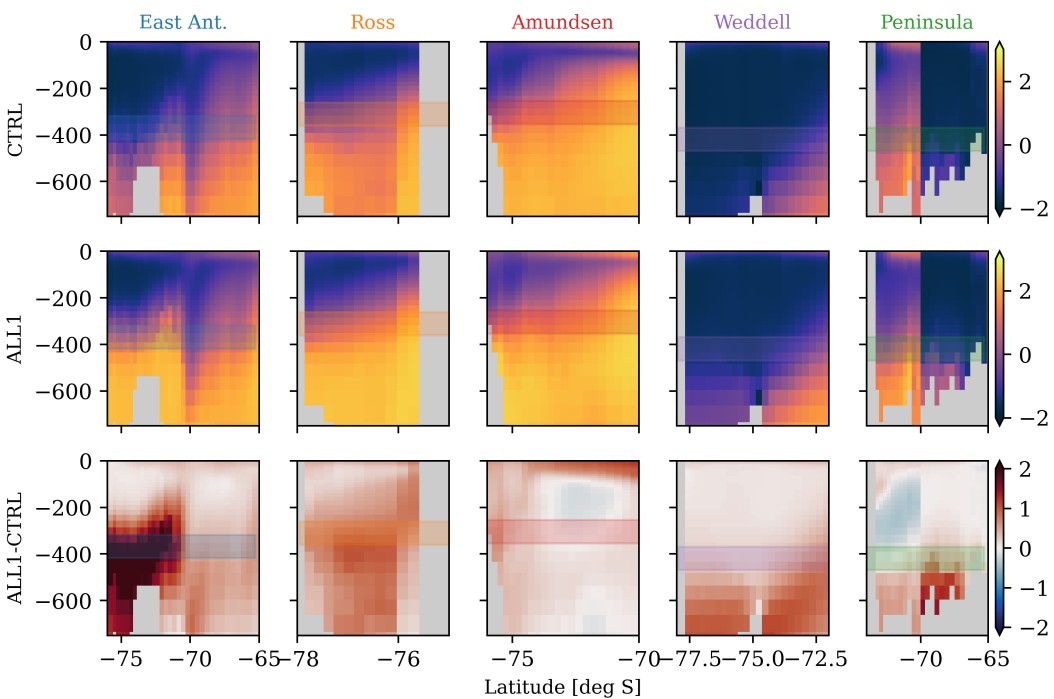

**Figure A2.** As Fig. 4, but for the ALL1 experiment (1000 Gt/yr), relative to the CTRL experiment.

## Appendix B: Ice mass Response Functions

In Fig. B1, we present the ice mass response functions from a subset of the LARMIP-2 ice sheet model ensemble, provided by the individual LARMIP-2 contributors.

## Appendix C: Earth System Model metrics

In order to assess the impact of historical temperature biases on the quantified meltwater–warming feedback, we compare various bulk metrics from our analysis between EC-Earth3 and the other ESMs. As shown by Purich and England (2021), a positive relation exists between the historical temperature and the future warming; in our analysis, this is the case for East Antarctica and the Ross region . EC-Earth3 has a relatively warm Ross and Amundsen Sea, and a cold Weddell Sea, as shown in Fig. C1. The regional warming is relatively strong in the East Antarctic, Weddell, and Peninsula regions. The calibrated $\gamma_c$ with feedback is lowest for EC-Earth3, likely relating to a relatively strong historical warming, which suppresses the meltwater–warming feedback for this model. In terms of the sea-level rise enhancement due to the feedback, EC-Earth3 falls in the middle of the pack and close to the ensemble median values of +5% and +80% as reported in the main text (Fig. 7 and 10). Note that

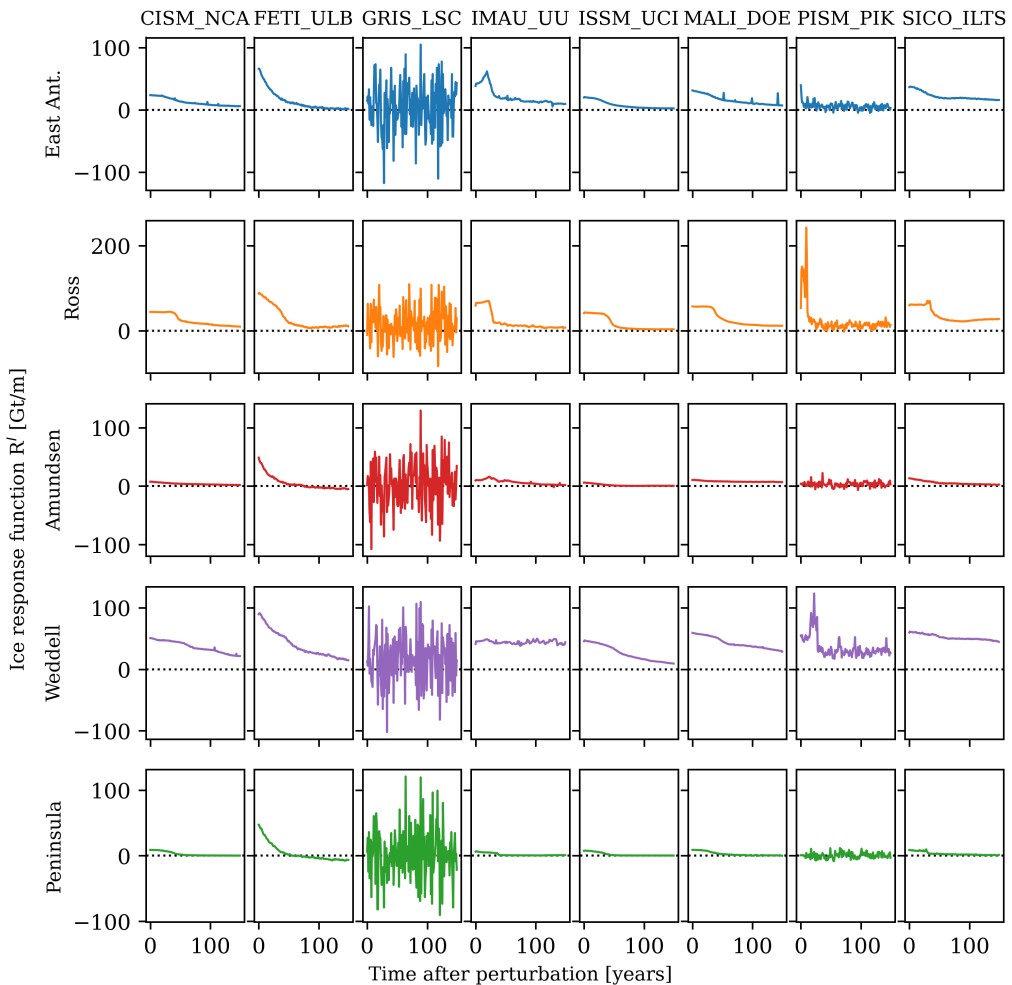

**Figure B1.** Ice mass loss response functions (IRFs) from 8 ice sheet models. The graphs denote $R_I$, indicating the mass loss rate $\dot{I}$ in response to a stepwise increase in the basal melt rate as defined in Eq. 4. The ice sheet models are, from left to right: CISM, FETISH, GRISLI, IMAU-ice, ISSM, MALI, PISM, and SICOPOLIS. Note the different y-scales for the different regions.

this analysis does not exclude the possibility that EC-Earth3 is more or less sensitive to meltwater-forcing, causing biases in the ORF. For this kind of intercomparison, we refer to the SOFIA initiative (Swart et al., 2023).

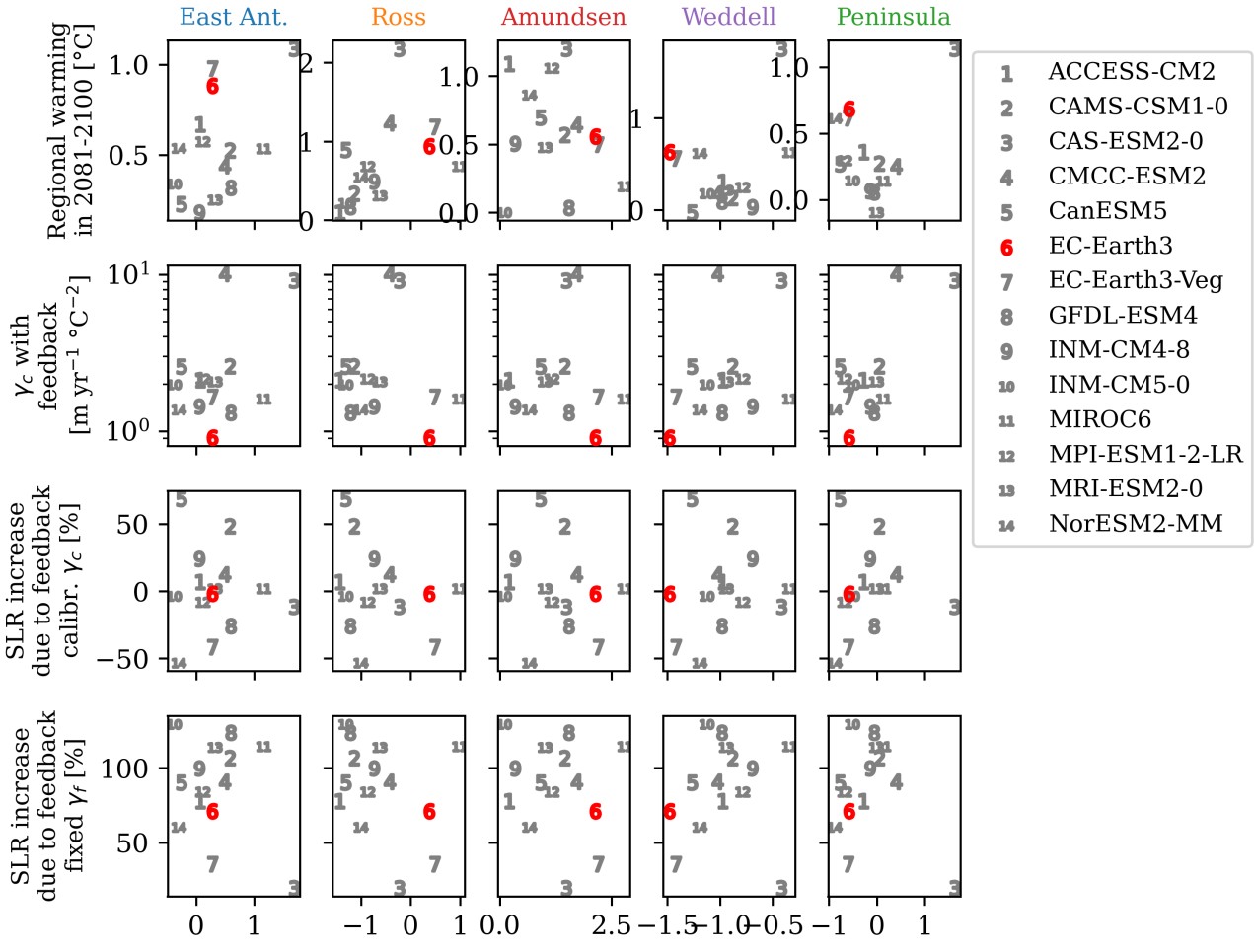

**Figure C1.** Bulk metrics of the 14 ESMs from which ocean temperature input is taken. Four metrics are shown as a function of the regional historical subsurface ocean temperature: the regional warming (top line, different per ocean region); the calibrated basal melt parameter $\gamma_c$ with feedback; the SLR increase in 2100 (feedback compared to no feedback) with the calibrated $\gamma_c$; and the same for the fixed $\gamma_f$. EC-Earth3 is highlighted as the red number 6. All values are based on SSP 2-4.5, and averaged over the ISMs where applicable.

**Appendix D: Sensitivity to ocean region**

The extent of the oceanic basins (Fig. 8c) includes both the continental shelf and the deep ocean. Various studies (Thompson et al., 2018; Beadling et al., 2022) have indicated that the oceanic warming in CMIP models and the oceanic response to

freshwater input differs across the continental shelf break. In addition, Hattermann and Levermann (2010) indicate that the quadratic basal melt parameterisation is applicable only to oceanic temperatures over the continental shelf. To assess the impact of this methodological choice (including ocean temperatures over the deep ocean), we have performed a sensitivity analysis. Here, both the ocean warming, derived from the 14 ESMs, and the ORFs are determined over the continental shelf only, defined by the 1000m isobath, following Barthel et al. (2020). The ocean warming in the ESMs overall is found to be slightly lower. However, the timescale of the ORFs is also somewhat shorter, whilst maintaining the magnitude of the response, resulting in a slightly stronger meltwater–warming feedback. Together, the impact of the horizontal extent of the oceanic basins is relatively small. In terms of the major quantitative results of this study:

- Using the fixed basal melt parameter $\gamma_f$, the median projected sea-level in 2100 increases by 110%, rather than 80%, for the three SSP scenarios.

- The calibrated basal melt parameter $\gamma_c$ reduces to 1.6 m yr$^{-1}$ °C$^{-2}$ due to the meltwater–warming feedback, rather than 1.7 m yr$^{-1}$ °C$^{-2}$.

- Using this calibrated basal melt parameter, the median projected sea-level in 2100 increases by 7%, rather than 5%, for the three SSP scenarios.

Altogether, the results of this study are qualitatively insensitive to the extent of the oceanic basins. Quantitatively, for EC-Earth3, the impact of the meltwater–warming feedback is slightly stronger when only considering the subsurface temperatures over the continental shelf region.

*Code and data availability.*    – Sea-level response functions from LARMIP-2 (Levermann et al., 2020): https://github.com/ALevermann/
Larmip2020/tree/master/RFunctions

   – Processed data and analysis code: Lambert (2025)

*Author contributions.*  EL and DLB designed the study. DLB gathered the LARMIP-2 data. EvdL and AJ provided support for the EC-Earth simulations. EL performed the computations and prepared the manuscript with contributions from all authors.

*Competing interests.*  The authors declare that they have no conflict of interest.

*Acknowledgements.*  This publication was supported by the Knowledge Programme Sea Level Rise which received funding from the Dutch Ministry of Infrastructure and Water Management. EL was further supported by the HiRISE project from Netherlands Organization for Scientific Research (grant no. OCENW.GROOT.2019.091. The authors thank Anders Levermann and the LARMIP-2 community for supporting the ice mass loss respone functions.)

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
