# Peer review of "Quantifying the feedback between Antarctic meltwater release and subsurface Southern Ocean warming"

_EGUsphere, 2024_

## Referee Comment (RC1)

**Review of "Quantifying the feedback between Antarctic meltwater release and subsurface Southern Ocean warming" by Lambert et al.**

In this manuscript, Lambert and co-authors analyze the potential impact of meltwater-induced ocean warming feedbacks on Antarctic ice sheet melting and the resulting sea-level contributions. I really enjoyed reading this paper and was impressed that they have managed to distill a very complicated analysis down to a relatively easy-to-understand story. They use a novel method involving Linear Response Functions derived from ocean temperature anomalies in perturbation experiments with EC-Earth3. By applying these functions across a suite of ESMs, they evaluate how different ocean warming scenarios, both with and without the feedback effect, may influence basal melt rates (and hence sea level rise). This approach provides valuable insights into the interactions between oceanic and cryospheric processes, enhancing our understanding of how these systems influence each other. The manuscript is well-structured, clearly written, and supported by high-quality figures. The analysis is comprehensive, the results are presented clearly, and the conclusions appear robust. The topic is of significant importance, as it addresses critical uncertainties in projecting future Antarctic ice-ocean interactions. It is, therefore, fitting for the audience Earth System Dynamics and will make a valuable contribution to the literature. Additionally, the findings are timely and relevant as the scientific community prepares for ISMIP7 and CMIP7. I have a few comments and concerns that I would like the authors to address prior to publication. My main concern is the chosen horizontal extent for ocean temperature extraction and the need for more discussion on the uncertainty of basing the ORFs on a single ESM.

**General comments**

1. **ORFs based on a single ESM (EC-Earth3)**
   As the authors note in the discussion, a significant uncertainty and limitation of this study is that the ORFs are derived solely from the EC-Earth3 model's response to meltwater input. The main assumption here is that other ESMs would exhibit a similar response. I understand that this is the only way to do this study without running experiments with many models (something covered by the SOFIA initiative), and I completely agree with your approach. I do not suggest changing this. However, I would appreciate a more extensive discussion on this topic. Specifically, it is likely that different models exhibit varying magnitudes of subsurface warming, as highlighted by Swart et al. (2023) and the recent work by Chen et al. (2023, https://doi.org/10.1029/2023GL106492 ). Furthermore, the assertion that most studies consistently show subsurface warming may be somewhat overstated. For instance, while you reference Thomas et al. (2023) and their findings of regional cooling, you could also include Beadling et al. (2022, https://doi.org/10.1029/2021JC017608 ), who demonstrated cooling around Western Antarctica in response to meltwater input in the GFDL-CM4 model. These studies suggest that a universal warming response is not guaranteed. If the ORFs were constructed using models like HadGEM3 or GFDL-CM4, the outcomes would likely be different. For example, Beadling et al. (2022) propose a mechanism in high-resolution GFDL-CM4 where additional meltwater isolates the West Antarctic shelf from offshore warm waters, a process that may not be well-represented in EC-Earth3. I believe the uncertainty stemming from this limitation is under-communicated, particularly in the results and conclusion sections. I recommend emphasizing earlier in the manuscript (perhaps also in the abstract) that the results are contingent on the assumption that all ESMs respond similarly to EC-Earth3, and that this

assumption could significantly affect the findings. I also suggest adding "in EC-Earth3" to the following sentence in the abstract: "Increased meltwater release from five individual Antarctic ice sheet regions is found to unambiguously warm the subsurface Southern Ocean at centennial time scales"

2. **Choice of horizontal and vertical boundaries for ocean temperature extraction**
   I am concerned about the uncertainties associated with the large horizontal extent and limited vertical extent when extracting temperature data from the 3D fields. Specifically, I question the inclusion of water masses located far offshore and hence spatial averaging over an area where know there are large gradients in T/S. Additionally, the quadratic melt rate parameterization (Eq.1) is for near the ice front and not far field (e.g. see discussion in Hattermann et al. (2010, https://link.springer.com/article/10.1007/s00382-009-0643-3 ). Would it not be more appropriate to restrict the analysis to water masses that are directly on the shelf, particularly those south of the 1000m isobath, as suggested by Barthel et al. (2020, https://doi.org/10.5194/tc-14-855-2020 )? Including offshore anomalies may risk incorporating temperature changes that do not influence the shelf directly, which is supported by findings from Beadling et al. (2022). Have you conducted a sensitivity analysis to determine whether the inclusion of these offshore waters significantly affects the results? Additionally, the narrow vertical extent makes the findings particularly susceptible to vertical biases within the models, as indicated by the results presented in Figure 4. The depth of the temperature anomalies varies from model to model, which could introduce further uncertainty. Have you explored how different the results might be if a wider depth range on the shelf were included? These depth estimates are realistic and based on reality, but in the model world, they might not capture the right water masses. I acknowledge that redoing the experiments and analysis is too much, and I do not request this, also I am well aware that this is nicely consistent with Levermann et al. (2020), but especially the choice of horizontal regions is suboptimal and inconsistent with similar studies.

3. **The peninsula ocean region includes both east and west**
   Given the distinct dynamics and differing water masses between the western and eastern parts of the Antarctic Peninsula, I question whether a weighted average between the two regions is meaningful or realistic from an ocean perspective. I understand that the IRF is just one function for the peninsula which makes sense from an ice perspective, but the oceanic feedback is likely very different on either side. I suggest separating these and either include both the western and eastern parts of the peninsula separately or just the eastern part. This could be done by using the same IRF, but different ORFs.

4. **Quadratic relation between basal melt rates and thermal forcing**
   In this paper, you chose a relatively simple way of converting thermal forcing to basal melt rates, which I appreciate given the already large uncertainty. However, you mention that "basal melt representation can be improved significantly by taking into account a more appropriate extrapolation into ice shelf cavities (Jourdain et al., 2020) and a more sophisticated calculation of basal melting (e.g., Lambert et al., 2023)". Can you explain why you did not use either of these two methods? Also, could you speculate on how these different methods might affect your

results? This sentence warrants some more discussion. Additionally, I am left with uncertainty about whether I should trust the calibrated or fixed basal melt parameter more. Using the fixed parameter gives a large difference between feedback and no feedback, whereas the calibrated parameter gives only a tiny (5%) difference. Could you be more clear on what the take-home message should be for the reader? Is it correct to assume that the calibrated parameter is more realistic and that I, therefore, can conclude that the inclusion of the oceanic feedback does not matter that much for the future projections? Or have I misunderstood this?

Additionally, I am curious to see the historical period (Figure 8) also for fixed non-callibrated gamma. This comparison is interesting and warrants some more discussion.

**Specific comments**

Line 4          Here, you have the opportunity to explain some more context for non-specialists. I  miss the "why" in the abstract. Why is this important? Why should we care about this feedback that you are aiming to quantify? By highlighting that this feedback is currently ignored in most simulations, you underscore the novelty and necessity of this work. It is clear in the introduction but would be useful to highlight this more in the abstract as well.

Line 27         You could add here that it decreases deep convection

Line 41         "on average" a positive one

Line 78         How representative is EC-Earth3 compared to other ESMs in simulating the mean state of the Southern Ocean? While Figure 3 provides a basic comparison with temperature data from reanalysis, it would be beneficial to include more references or a detailed analysis that evaluates how well EC-Earth3 reproduces key aspects of Southern Ocean hydrography. How do its biases compare with those of other models? Especially important for this study is how well it captures the vertical structure and the Antarctic slope current in comparison to observations. Including an additional figure and some discussion to demonstrate that EC-Earth3 is not an outlier among ESMs would strengthen the case for basing the ORFs on this model.

Line 121        "from either one of the five source regions." This was a little unclear to me at first. You run 5 experiments, where you just change one region in each experiment and do nothing with the other regions. Can you make this more clear and also explain earlier on why you do not just change all regions in one single experiment?

Line 125        (and throughout the paper): For easier comparison with other studies, please consider including a comparison of the Sverdrup equivalent to "Gt/yr" in the methods section. (at least when describing the pertubution magnitudes (see Swart et al., 2023).

Line 141        Can you include a table of the ESMs in the Appendix?

Line 151        Why constant and why -1.7? At the depths you have chosen the freezing temp should be much lower. Not? Could you refine Tf for each region by calculating the freezing temperature based on the mean salinity from the EC-Earth piControl at the depths

where you compute the melt rate? This adjustment would likely make your ORF estimates more realistic.

Line 142    "For each ESM we use the piControl simulation to bias correct long-term ocean temperature trends". Please expand. This step is not clear. Why?

Line 252    The reanalysis data referenced here and utilized in Figure 3 should be described in greater detail within the methods section. It would be beneficial to clarify how these reanalysis products are combined and whether they have undergone any evaluation to assess their reliability. Why have these reanalysis products been chosen over observational climatologies (e.g. Jourdain et al., 2020)?

Line 276    This is a very interesting result and can be highlighted. Please expand a bit more on this.

Line 280    "coloured lines in Fig. 5." Please specify we have to look at the rightmost column.

Line 315    This is a nice example, and illustrated the methodology very well. It was absolutely needed fo me to understand it, so thanks for this. However, why did you choose CISM-NCA? This is unclear. Can you explain this choice or is it random?

Line 332    I understand the limitation with the fixed basal melt parameter. However, Figure 7 is still a key result, and I think it deserves some more text. For example, the differences between scenarios depicted in this figure warrant a more detailed description. If Fig. 7 is to be included, it deserves a bit more than two sentences in the result section.

Line 385    I am not entirely convinced by the assertion that the general quantification of the meltwater–warming feedback is robust. While it may hold for EC-Earth3, your own discussion suggests that if the ORFs were based on a different model, the feedback could vary significantly. Therefore, it is unclear how this conclusion can be generalized. I acknowledge that this estimate is likely the best possible given the presented analysis, but I would suggest modifying this statement to reflect the associated uncertainty more accurately.

Line 395    The last paragraph of the discussion is very good and very important. I suggest moving this to the conclusion instead.

Line 402    The improvement/difference in the freshwater balance from the original CMIP6 version of EC-Earth3 and the new CTRL with the modified masks and routing is a significant result. The difference from the new distribution is substantial, and I believe it warrants a sentence or two in the conclusions. It is important for other modeling centers as you have shown that the way freshwater is added can have almost equally much to say as to how much freshwater is added. This result should be highlighted more. (Perhaps also in the abstract if you have space).

Figure 3    Caption: state clearly in the first sentence that is for EC-Earth3 (helps the reader). e.g. "Control time series of subsurface ocean temperatures from EC-Earth3." Similar for Figures 4 and 5.

Figure 3    Caption: repeat here that the CTRL has the adjusted/improved freshwater balance.

| | |
|---|---|
| Figure 4 | It would greatly enhance the manuscript to include an additional figure, similar to the lower row in Figure 4 (or add rows to this figure) displaying the temperature anomalies to the 200 and 400 GT/yr perturbation experiment in EC-Earth. This will also provide valuable insight or confirm whether the choice of vertical/horizontal extent of temperature anomalies is good for EC-Earth3. |
| Figure 4 | I recommend adjusting the x-axis scales for each region to optimize data visibility. It is more important to ensure that the data is clearly represented than to maintain a uniform latitudinal extent across all plots. For example, the data for the Ross region is currently difficult to discern, and refining the scale could make the patterns more apparent and interpretable. |
| Figure 5 | Caption: please add "Note different y-axis scales". |
| Figure 6 | Panels are very small and hard to read. Consider whether a different y-axis may be appropriate, especially for the second and third rows. (could, for example, have them all similar with the exception of the last column) Also, make sure that this Figure fills the full width of the page. Currently, the difference between feedback and no feedback is very hard to see. |
| Figure 8 | Just EC-Earth? And add that this is for all regions combined? |
| Figure 9 | These are the same for all regions, right? Perhaps repeat that in the caption. |
| Figure 10 | It would be nice to include the median from Figure 7, the fixed basal melt parameter, on this figure for comparison. |
| Figure A1 | This figure is useful for the community, but I would suggest a different y-axis for each region. The main point of this figure (that the response is the same for all experiments) still remains clear. The same is true for Figure B1. |

---

## Author Comment (AC1)

**Review of "Quantifying the feedback between Antarctic meltwater release and subsurface Southern Ocean warming" by Lambert et al.**

In this manuscript, Lambert and co-authors analyze the potential impact of meltwater-induced ocean warming feedbacks on Antarctic ice sheet melting and the resulting sea-level contributions. I really enjoyed reading this paper and was impressed that they have managed to distill a very complicated analysis down to a relatively easy-to-understand story. They use a novel method involving Linear Response Functions derived from ocean temperature anomalies in perturbation experiments with EC-Earth3. By applying these functions across a suite of ESMs, they evaluate how different ocean warming scenarios, both with and without the feedback effect, may influence basal melt rates (and hence sea level rise). This approach provides valuable insights into the interactions between oceanic and cryospheric processes, enhancing our understanding of how these systems influence each other. The manuscript is well-structured, clearly written, and supported by high-quality figures. The analysis is comprehensive, the results are presented clearly, and the conclusions appear robust. The topic is of significant importance, as it addresses critical uncertainties in projecting future Antarctic ice-ocean interactions. It is, therefore, fitting for the audience Earth System Dynamics and will make a valuable contribution to the literature. Additionally, the findings are timely and relevant as the scientific community prepares for ISMIP7 and CMIP7. I have a few comments and concerns that I would like the authors to address prior to publication. My main concern is the chosen horizontal extent for ocean temperature extraction and the need for more discussion on the uncertainty of basing the ORFs on a single ESM.

*Thank you for this positive and constructive review. Please find our point-by-point replies below in blue.*

**General comments**

1.      **ORFs based on a single ESM (EC-Earth3)** As the authors note in the discussion, a significant uncertainty and limitation of this study is that the ORFs are derived solely from the EC-Earth3 model's response to meltwater input. The main assumption here is that other ESMs would exhibit a similar response. I understand that this is the only way to do this study without running experiments with many models (something covered by the SOFIA initiative), and I completely agree with your approach. I do not suggest changing this. However, I would appreciate a more extensive discussion on this topic. Specifically, it is likely that different models exhibit varying magnitudes of subsurface warming, as highlighted by Swart et al. (2023) and the recent work by Chen et al. (2023, https://doi.org/10.1029/2023GL106492 ). Furthermore, the assertion that most studies consistently show subsurface warming may be somewhat overstated. For instance, while you reference Thomas et al. (2023) and their findings of regional cooling, you could also include Beadling et al. (2022, https://doi.org/10.1029/2021JC017608 ), who demonstrated cooling around Western Antarctica in response to meltwater input in the GFDL-CM4 model. These studies suggest that a universal warming response is not guaranteed. If the ORFs were constructed using models like HadGEM3 or GFDL-CM4, the outcomes would likely be different. For example, Beadling et al. (2022) propose a mechanism in high-resolution GFDL-CM4 where additional meltwater isolates the West Antarctic shelf from offshore warm waters, a process that may not be well-represented in EC-Earth3. I believe the uncertainty stemming from this limitation is under-communicated, particularly in the results and conclusion sections. I recommend emphasizing earlier in the manuscript (perhaps also in the abstract) that the results are contingent on the assumption that all ESMs respond similarly to EC-Earth3, and that this assumption could significantly affect the findings. I also suggest adding "in EC-Earth3" to the following sentence in the abstract: "Increased meltwater release from five individual Antarctic ice sheet regions is found to unambiguously warm the subsurface Southern Ocean at centennial time scales"

We agree that the current inter-model uncertainty in the response to freshwater forcing (and hence the resultant ORFs) is significant, both in terms of magnitude and sign. We are glad to read that the reviewer agrees that an inter-model comparison is beyond the scope of this study and is effectively addressed by the SOFIA initiative. We will therefore follow the reviewer's suggestions, emphasising more strongly the dependence of our results on the choice of ESM, and we will expand the discussion on published inter-model disagreements and their impact on our results and conclusions.

2.     **Choice of horizontal and vertical boundaries for ocean temperature extraction** I am concerned about the uncertainties associated with the large horizontal extent and limited vertical extent when extracting temperature data from the 3D fields. Specifically, I question the inclusion of water masses located far offshore and hence spatial averaging over an area where know there are large gradients in T/S. Additionally, the quadratic melt rate parameterization (Eq.1) is for near the ice front and not far field (e.g. see discussion in Hattermann et al. (2010, https://link.springer.com/article/10.1007/s00382-009-0643-3 ). Would it not be more appropriate to restrict the analysis to water masses that are directly on the shelf, particularly those south of the 1000m isobath, as suggested by Barthel et al. (2020, https://doi.org/10.5194/tc-14-855-2020 )? Including offshore anomalies may risk incorporating temperature changes that do not influence the shelf directly, which is supported by findings from Beadling et al. (2022). Have you conducted a sensitivity analysis to determine whether the inclusion of these offshore waters significantly affects the results? Additionally, the narrow vertical extent makes the findings particularly susceptible to vertical biases within the models, as indicated by the results presented in Figure 4. The depth of the temperature anomalies varies from model to model, which could introduce further uncertainty. Have you explored how different the results might be if a wider depth range on the shelf were included? These depth estimates are realistic and based on reality, but in the model world, they might not capture the right water masses. I acknowledge that redoing the experiments and analysis is too much, and I do not request this, also I am well aware that this is nicely consistent with Levermann et al. (2020), but especially the choice of horizontal regions is suboptimal and inconsistent with similar studies.

We agree that this methodological choice introduces an uncertainty that is not quantified explicitly. To address this concern, we will perform a sensitivity analysis by masking the ocean regions deeper than 1000m and extending the vertical extent to the original depth range of Levermann et al 2020. As we believe the manuscript at present to already be quite extensive, we propose to include this sensitivity analysis in the Appendix in the form of a reproduction of Fig 7 and/or 10. Dependent on the resultant sensitivity, we will reflect on this in the main text.

3.     **The peninsula ocean region includes both east and west**
Given the distinct dynamics and differing water masses between the western and eastern parts of the Antarctic Peninsula, I question whether a weighted average between the two regions is meaningful or realistic from an ocean perspective. I understand that the IRF is just one function for the peninsula which makes sense from an ice perspective, but the oceanic feedback is likely very different on either side. I suggest separating these and either include both the western and eastern parts of the peninsula separately or just the eastern part. This could be done by using the same IRF, but different ORFs.

We agree that the oceanic feedback is likely different on either side of the Peninsula. However, since we have only one IRF for this region, we cannot separate these ocean/ice feedbacks. New ice sheet model experiments would be required with basal melt perturbations on either side. Because the Peninsula contributes minimally to both total ice mass loss and the sea-level response, we consider this region of minimal interest for the purpose of our study. This is best reflected in Fig 6. Although we agree that this region is oceanographically interesting, the minimal contribution to the total feedback

– in our eyes – does not validate additional analysis or focus on this specific region. We will explain this in the manuscript where we introduce the Peninsula region.

4. **Quadratic relation between basal melt rates and thermal forcing**

In this paper, you chose a relatively simple way of converting thermal forcing to basal melt rates, which I appreciate given the already large uncertainty. However, you mention that "basal melt representation can be improved significantly by taking into account a more appropriate extrapolation into ice shelf cavities (Jourdain et al., 2020) and a more sophisticated calculation of basal melting (e.g., Lambert et al., 2023)". Can you explain why you did not use either of these two methods? Also, could you speculate on how these different methods might affect your results? This sentence warrants some more discussion. Additionally, I am left with uncertainty about whether I should trust the calibrated or fixed basal melt parameter more. Using the fixed parameter gives a large difference between feedback and no feedback, whereas the calibrated parameter gives only a tiny (5%) difference. Could you be more clear on what the take-home message should be for the reader? Is it correct to assume that the calibrated parameter is more realistic and that I, therefore, can conclude that the inclusion of the oceanic feedback does not matter that much for the future projections? Or have I misunderstood this? Additionally, I am curious to see the historical period (Figure 8) also for fixed non-callibrated gamma. This comparison is interesting and warrants some more discussion.

The simple choice of a uniform basal melt sensitivity is constrained by the original LARMIP-2 methodology and a more realistic representation would require redoing the ice sheet model experiments from which the LRFs are constructed. We will clarify this statement.
Regarding the interpretation of the calibrated vs fixed gamma's, the relevant take-home message depends on the reader. We will elaborate on this interpretation.

**Specific comments**

Line 4 Here, you have the opportunity to explain some more context for non-specialists. I miss the "why" in the abstract. Why is this important? Why should we care about this feedback that you are aiming to quantify? By highlighting that this feedback is currently ignored in most simulations, you underscore the novelty and necessity of this work. It is clear in the introduction but would be useful to highlight this more in the abstract as well.

Agreed, we will state the importance more clearly in the abstract

Line 27 You could add here that it decreases deep convection

Agreed, we will include this

Line 41 "on average" a positive one

Agreed, will implement

Line 78 How representative is EC-Earth3 compared to other ESMs in simulating the mean state of the Southern Ocean? While Figure 3 provides a basic comparison with temperature data from reanalysis, it would be beneficial to include more references or a detailed analysis that evaluates how well EC-Earth3 reproduces key aspects of Southern Ocean hydrography. How do its biases compare with those of other models? Especially important for this study is how well it captures the vertical structure and the Antarctic slope current in comparison to observations. Including an additional figure and some discussion to demonstrate that EC-Earth3 is not an outlier among ESMs would strengthen the case for basing the ORFs on this model.

Based on the intercomparison of Heuze (https://doi.org/10.5194/os-17-59-2021), EC-Earth3 is not an outlier in terms of Southern Ocean indicators despite its warm bias. We will refer to this study and point this out explicitly while introducing EC-Earth3. In addition, we will include a quantitative inter-model comparison as requested by the other reviewer, which includes EC-Earth3.

Line 121 "from either one of the five source regions." This was a little unclear to me at first. You run 5 experiments, where you just change one region in each experiment and do nothing with the other regions. Can you make this more clear and also explain earlier on why you do not just change all regions in one single experiment?

Yes, we will make this more clear.

Line 125 (and throughout the paper): For easier comparison with other studies, please consider including a comparison of the Sverdrup equivalent to "Gt/yr" in the methods section. (at least when describing the pertubution magnitudes (see Swart et al., 2023).

This is a good suggestion, we will include this.

Line 141 Can you include a table of the ESMs in the Appendix?

Yes we will include this

Line 151 Why constant and why -1.7? At the depths you have chosen the freezing temp should be much lower. Not? Could you refine Tf for each region by calculating the freezing temperature based on the mean salinity from the EC-Earth piControl at the depths where you compute the melt rate? This adjustment would likely make your ORF estimates more realistic.

The ORFs are independent of this basal melt formulation (see Fig 2b). In addition, albeit highly idealised, the exact value of Tf is of second order influence and negligible in respect to other sources of uncertainty. We therefore opted for simplicity and transparency over accuracy and complexity. We will state this more clearly in the text.

Line 142 "For each ESM we use the piControl simulation to bias correct long-term ocean temperature trends". Please expand. This step is not clear. Why?

This was indeed an unclear formulation. We simply detrend the temperatures to remove long-term drift. We will state this more clearly in the text.

Line 252 The reanalysis data referenced here and utilized in Figure 3 should be described in greater detail within the methods section. It would be beneficial to clarify how these reanalysis products are combined and whether they have undergone any evaluation to assess their reliability. Why have these reanalysis products been chosen over observational climatologies (e.g. Jourdain et al., 2020)?

As our work builds upon that of van der Linden et al (2023), we have used the same reference values as theirs. Here we have prioritised consistency with this work for optimal comparison and will be more explicit on this in the text. In addition, we will include a brief description of the underlying products.

Line 276 This is a very interesting result and can be highlighted. Please expand a bit more on this.

We agree that this is interesting and will expand somewhat. However, the reviewer pointed out correctly, results like these are likely dependent on the ESM of choice, and a detailed analysis of the

underlying mechanisms is beyond the scope of this study. Hence, this expansion will be based on interpretation rather than in-depth analysis.

Line 280 "coloured lines in Fig. 5." Please specify we have to look at the rightmost column.

Agreed, will do

Line 315 This is a nice example, and illustrated the methodology very well. It was absolutely needed fo me to understand it, so thanks for this. However, why did you choose CISM-NCA? This is unclear. Can you explain this choice or is it random?

We will explain that this is a quasi-random choice of a representative ISM-ESM set.

Line 332 I understand the limitation with the fixed basal melt parameter. However, Figure 7 is still a key result, and I think it deserves some more text. For example, the differences between scenarios depicted in this figure warrant a more detailed description. If Fig. 7 is to be included, it deserves a bit more than two sentences in the result section.

We agree that the description of the results in Fig 7 was overly brief and will expand this.

Line 385 I am not entirely convinced by the assertion that the general quantification of the meltwater–warming feedback is robust. While it may hold for EC-Earth3, your own discussion suggests that if the ORFs were based on a different model, the feedback could vary significantly. Therefore, it is unclear how this conclusion can be generalized. I acknowledge that this estimate is likely the best possible given the presented analysis, but I would suggest modifying this statement to reflect the associated uncertainty more accurately.

We agree that this statement requires more nuance and will reformulate it accordingly.

Line 395 The last paragraph of the discussion is very good and very important. I suggest moving this to the conclusion instead.

This is a good suggestion and in line with the other reviewer's comment that the Conclusion section is too brief. We will adopt this.

Line 402 The improvement/difference in the freshwater balance from the original CMIP6 version of EC-Earth3 and the new CTRL with the modified masks and routing is a significant result. The difference from the new distribution is substantial, and I believe it warrants a sentence or two in the conclusions. It is important for other modeling centers as you have shown that the way freshwater is added can have almost equally much to say as to how much freshwater is added. This result should be highlighted more. (Perhaps also in the abstract if you have space).

We agree that this is interesting and possibly important. However, we have omitted this from the conclusions as we cannot prove that the improvement is 'for the right reasons' (and, again, to what extent this applies to other ESMs). We will include a sentence on this in the conclusions with the necessary caveat.

Figure 3 Caption: state clearly in the first sentence that is for EC-Earth3 (helps the reader). e.g. "Control time series of subsurface ocean temperatures from EC-Earth3." Similar for Figures 4 and 5.

Agreed, we will include this.

Figure 3 Caption: repeat here that the CTRL has the adjusted/improved freshwater balance.

Agreed, we will include this

Figure 4 It would greatly enhance the manuscript to include an additional figure, similar to the lower row in Figure 4 (or add rows to this figure) displaying the temperature anomalies to the 200 and 400 GT/yr perturbation experiment in EC-Earth. This will also provide valuable insight or confirm whether the choice of vertical/horizontal extent of temperature anomalies is good for EC-Earth3.

We will explore adding a row below this figure. Alternatively, we will add a figure to the appendix to visualise the response in the ALL1 experiment.

Figure 4 I recommend adjusting the x-axis scales for each region to optimize data visibility. It is more important to ensure that the data is clearly represented than to maintain a uniform latitudinal extent across all plots. For example, the data for the Ross region is currently difficult to discern, and refining the scale could make the patterns more apparent and interpretable.

We agree and will modify the x-axis scales.

Figure 5 Caption: please add "Note different y-axis scales".

Agreed and will implement

Figure 6 Panels are very small and hard to read. Consider whether a different y-axis may be appropriate, especially for the second and third rows. (could, for example, have them all similar with the exception of the last column) Also, make sure that this Figure fills the full width of the page. Currently, the difference between feedback and no feedback is very hard to see.

We will adjust this figure to full width and will optimise visibility. However, part of the message of this figure is that, in the Amundsen and Peninsula regions, the difference is indiscernible. So we will not zoom in on these insignificant differences.

Figure 8 Just EC-Earth? And add that this is for all regions combined?

Agreed, we will include this

Figure 9 These are the same for all regions, right? Perhaps repeat that in the caption.

Agreed, we will include this

Figure 10 It would be nice to include the median from Figure 7, the fixed basal melt parameter, on this figure for comparison.

We believe this will include too much confusion and prefer keeping a strict division between the fixed and calibrated gamma's. Instead we will add in the caption that the 'no feedback' lines are identical in Figs 7 and 10, which serves as a reference.

Figure A1 This figure is useful for the community, but I would suggest a different y-axis for each region. The main point of this figure (that the response is the same for all experiments) still remains clear. The same is true for Figure B1.

Agreed, we will implement this.

---

## Author Comment (AC2)

The manuscript by Lambert and colleagues uses output from Earth system models, ice sheet models, and observations to estimate the impact of an ocean-ice feedback on sea level projections. The majority of Earth system models do not yet include coupled ice sheets and ice shelves, and thus ice-ocean feedbacks are not currently included in sea level projections. Additional meltwater from Antarctica increases stratification, causing surface cooling and subsurface warming at the depth of ice shelf basal melting, which has the potential to accelerate melting. Making use of existing model output, the authors calculate linear response functions to estimate the impact of this feedback on sea level projections. They find at first pass, that this feedback could increase sea level projections for 2100 by 80%, which is a substantial effect. However, they also note that the basal melt parameter for this estimation has not been calibrated on observed historical melt rates. When accounting for calibration, they revise their estimate to suggest that the feedback could increase sea level projections for 2100 by 5%.

The findings of this manuscript will be useful to the scientific community working towards coupling dynamic ice sheets and shelves into Earth system models. The manuscript is well written, and the methodology is clear described. I have several minor comments specified below, which I expect will be straightforward to address.

Thank you for your positive and constructive review. Please find our point-by-point replies below in blue.

I also have one major comment, regarding bias / potential bias in the Earth system model ocean component that will influence the ocean response function and results – sensitivity to this should be investigated and presented – currently, it is not clear how dependent the results are too possible ocean-model bias in both the mean state and trends. I thus recommend major revisions for this manuscript.

We agree that ESM biases provide a significant uncertainty in our results. This comes in two forms: 1) in the ORFs derived from EC-Earth3, and 2) in the historical and future warming from a suite of ESMs.

Point 1 was also raised by the other reviewer, who acknowledged that an explicit sensitivity analysis would require an intercomparison study; this is the very purpose of the SOFIA initiate and hence beyond the scope of this study. To address this point, we will elaborate on the biases in EC-Earth3 with respect to other ESMs; we will discuss in more depth the response to increased meltwater forcing in contrast to other ESMs; and we will add a clear disclaimer that our results are dependent on a single ESM, including in the abstract.

Point 2 will be addressed with a quantitative inter-model assessment as described below where this point is raised explicitly. We hope that these proposed modifications satisfy the reviewer's concerns.

Specific comments

Abstract line 10-14: The three sentences with reported percentages read a bit as unconnected listed items, and this has the effect of making the percentages seem unrelated. Rewording the calibration methodology and revised sea level enhanced projection sentences would help here.

We agree and will rephrase to better highlight these results.

Line 40: "SOFIAMIP project" would be better described as "Southern Ocean Freshwater Input from Antarctica (SOFIA) Imitative".

Agreed, we will implement this

Line 40-42: Can you add citations for these modelling studies in the last sentence – as it is currently written, this seems like it is referring to the SOFIA model output from Swart et al. (2023), which is not the case.

Yes, we will add these

Line 50-51: Do any of models or the multimodel mean agree with the observations?

Individual ensembles of individual models do, but not the multimodel mean. Because there is an intense debate in the ice sheet community (and the design of ISMIP7) on this topic, we'd prefer to stay away from this in the present study. Hence, we will not elaborate on this further.

Line 89: Suggest "redistributed" here.

Agreed, will implement

Line 99-102: It is implicit here that the Antarctic runoff generated in the CMIP6 version of the model is no longer generated in this version, but it would be helpful to describe this explicitly, so that it is clear you aren't double counting this runoff plus your more realistic representation of meltwater release.

Good point, we will state this explicitly.

Line 142: Does "bias correct" mean bias correct the mean state, or to bias correct the trends, i.e. de-drift? I think it means the latter. How is this bias correction / de-drifting done? E.g. have you removed linear grid point trends from the piControl in the corresponding historical and SSP scenarios?

Yes, this point was unclear. We refer here to detrending rather than bias correcting. We will rephrase this to clarify.

Further, the historical CMIP6 ocean trends around the Antarctic margins themselves are biased (Purich and England 2021) – have you corrected for this? For example, multimodel mean historical trends around the margins show more warming than observed in most regions, except the Amundsen Sea, over 1975-2012. The historical ocean trends will presumably exert a strong influence on your projections and gamma calibration. Sensitivity to model bias in both the mean-state and trends should be investigated and reported in the manuscript.

We agree that the aspect of model bias deserves more attention in our study. A bias correction to historical trends is unfeasible due to the large uncertainties in observations (Schmidtko et al., 2014). However, Purich and England (2021) indeed reveal an important positive relationship between historical biases and future trends. To investigate the relationship between

ESM bias, future warming, gamma calibration, and sea-level contribution, we will include either a table or a figure with inter-model correlations in the Appendix.

Purich, A., & England, M. H. (2021). Historical and future projected warming of Antarctic Shelf Bottom Water in CMIP6 models. Geophysical Research Letters, 48, e2021GL092752. https://doi.org/10.1029/2021GL092752

Line 169: Computation.

Thank you, will correct.

Line 204-207: It would be useful if you list the ice sheet models and ESMs used in this study in a table.

We agree and will list these in a table in the appendix.

Line 220-222: It would be useful here to specify that you are using the observed historical ice mass loss, and the modelled ocean warming. What is the uncertainty in the historical ice mass loss? What is the uncertainty in the ocean warming?

We will elaborate on this.

Line 232-235: What does it mean that some model pairs require a negative gamma to reproduce the positive sea level contribution? Is the model representation of ocean temperature around the Antarctic margins suitable for this use?

No they are not and are therefore excluded. We will explain this in more detail.

Fig. 4 caption: Capitalisation should be "western Peninsula" and "eastern Peninsula"

Thank you, we will correct.

Line 275: In this section it would be useful here to comment on this subsurface warming relative to other studies (particular in the Amundsen Sea), and comment on the resolution of this model / limitations to representation of high-resolution processes that may affect the response to meltwater.

We agree and will expand on this here.

Line 293-295: This saturation effect has been reported previously, e.g. in Schloesser et al. (2019).

Thank you for pointing this out, we will cite this study accordingly.

Schloesser, F., T. Friedrich, A. Timmermann, R. M. DeConto, and D. Pollard, 2019: Antarctic iceberg impacts on future Southern Hemisphere climate. Nat. Climate Change, 9, 672– 677, https://doi.org/10.1038/s41558-019-0546-1.

Line 296-298: The upper limit seems reasonable. Did you experiment with other (non-linear) fits?

Yes, we have explored a non-linear fit. However, the sensitivity of the final results to this choice was small and did not validate the introduction of a more complex mathematical description. We will briefly mention this in the text.

Line 319: Specify "subsurface" warming.

Agreed, will do

Line 353-354: Specify "the last decades of our projections", so it is clear that you are not referring to the last decades of the observations. Also, here and elsewhere, the date that you are presenting projections for (i.e. 2100) should be specified.

Agreed, we will specify this.

Line 363: Specify "surface" cooling.

Agreed, will do

Line 364-365: Can you elaborate on why it could lead to an overestimation of sea-level projections?

Yes, we will elaborate on this

Line 375-377: Have you shown the warming around Antarctica in EC-Earth3, compared to other models? Can you support this statement with references?

The references are included in the following sentences. We will reorganise this paragraph to avoid confusion.

Line 380: "Contrast"

Thank you, will correct

Conclusions: Some of the conclusions text seems like it is written in brief draft format. I recommend revising the text.

We agree. A similar comment was made by the other reviewer. To address this, we will move the last paragraph of the Discussion to the Conclusions and expand the Conclusions overall.

---

## Author Response (AR2)

Dear editor, please find below our point-by-point reply to both reviewer's comments in green.

**Reviewer 1**

Dear Editor and Authors,

Thank you for sending the revised manuscript and the point-by-point responses to my comments. I enjoyed reading the manuscript again, and it is still a very nice paper that the community will value tremendously.

I have reviewed the revisions and am satisfied with the changes made. The authors have addressed all of my concerns appropriately, and I believe the manuscript has improved. I am happy to recommend the manuscript for publication pending a few minor revisions (see below).

We thank the reviewer for yet another review and their useful comments. Please find our replies below.

Line 437 (in the tracked-changes document): I am happy with the added discussion here. But: "An inter-model comparison within the SOFIA initiative (Swart et al., 2023) showed a qualitative agreement on subsurface warming along the Antarctic coast (Chen et al., 2023)." This is not correct and needs to be changed/removed. Chen et al. show a qualitative agreement on subsurface warming in the zonal mean. This does not mean that the models show a warming along the coast. Current work in prep in the SOFIA community shows that this "zonal mean" warming is dominated by certain regions and that there is cooling along the coast in some regions in some models. This work is, sadly, not yet citable. I would either remove this sentence completely or state that a recent intermodel comparison shows that, in the zonal average, models agree on a subsurface warming but that regional differences are the focus of current investigations, such as those within the SOFIA initiative.

We agree that this statement should clarify the zonal mean aspect of the results of Chen et al. As we cannot cite these unpublished results, and

removal of this sentence would require removing a large substance of this discussion paragraph, we have retained the sentence and stress that this regards zonal mean changes.

Regarding the constant freezing temperature, -1.7. I understand your wish for simplicity and transparency, and I see that rerunning the experiments with different values would be a lot of work for likely a very small change. The added argumentation in the methods section is an improvement, but I am not yet fully satisfied with it. Firstly, you state that you have chosen the surface freezing temperature, correct, but I think you need to add that this an unrealistic value and that the freezing temperature at depth is closer to –2.0 - -2.3 (for your depth ranges). Secondly, I think you need to be more quantitative and precise in the following statement: "We consider the uncertainty stemming from this idealization to be minimal in comparison to uncertainties from other methodological assumptions." What is minimal? What difference approximately can we expect if we use -1.7 instead of -2.3? Can you provide an order of magnitude? And can you be more precise on which other assumptions you refer to her?

We agree that the choice of freezing temperature is in hindsight not ideal, but also agree with the reviewer that at this stage, new simulations are undesirable. Without a sensitivity experiment, we cannot provide a quantitative assessment of this uncertainty however. Instead, we have more explicitly stated that lower freezing temperatures would be more appropriate, and have explicitly formulated the methodological choices we consider to contribute (much) more to the overall uncertainty and idealisation of this study.

Gt/yr vs. Sv: I see you've noted that 1 Gt/yr = 0.0317 mSv, which is helpful. However, to improve clarity for the reader, I recommend consistently including the equivalent value in Sverdrups whenever you present a value in Gt/yr. This will save readers from having to do the conversion themselves each time. For instance, "an increase in ice mass loss of 400 Gt/yr (≈0.012 Sv)".

We have moved the conversion factor to the first point where the unit Gt/yr is introduced. Additionally, for all unique values cited (200, 400, 1000, 2000, and 3300) we have added the conversion to Sv wherever these values is first introduced. We have not included this every time any value in Gt/yr is mentioned however, as this would limit the readability of our manuscript.

**Reviewer 2**

Second review of "Quantifying the feedback between Antarctic meltwater release and subsurface Southern Ocean warming" by Lambert et al.

In their revision, the authors have suitable addressed my comments from the first review. The revised manuscript makes it clear where results are based on a single coupled model and includes an expanded discussion on biases in EC-Earth. Where practicable for this study, a comparison of results across coupled models are presented. The findings of this manuscript will be useful to the scientific community working towards coupling dynamic ice sheets and shelves into Earth system models, and I consider this manuscript in its revised form suitable for publication in ESD.

We thank the reviewer for another review. Please find below our replies.

I have two very minor comments for consideration:

Line 159-160 (of tracked changes version): I appreciate the clarification regarding dedrifting. It would be useful to briefly include mention of the period over which the piControl linear trend was calculated, e.g.100 years? The full available piControl length?

The dedrifiting is based on the full piControl. We have included this in the manuscript.

Line 257-260 (of tracked changes version): The revised text makes your methodology and reasoning clearer, but I wonder, for all regions, is regional net cooling over the historical period actually unrealistic? Over shorter periods (1975-2012) there is cooling in the Weddell and Ross (Schmidtko et al. 2014), which may be due to natural variability, a forced change causing the cooling, or sparse observations not fully sampling the real change. I don't expect you to change your methodology here, but I think your phrasing of "unrealistic" needs refining, i.e. "which we consider to be unrealistic over a 150-year period with increasing global temperatures" or something like that – then it is explicit why you assume it to be unrealistic.

Indeed, a historical cooling is certainly plausible in specific regions. What we referred to as unrealistic is the negative basal melt sensitivity that results from this. We have clarified this in the text.